# Improving Few-Shot Generalization by Exploring and Exploiting Auxiliary Data

**Alon Albalak**
University of California,
Santa Barbara
alon_albalak@ucsb.edu

**Colin Raffel**
University of Toronto
Vector Institute
craffel@gmail.com

**William Yang Wang**
University of California,
Santa Barbara
william@cs.ucsb.edu

## Abstract

Few-shot learning is valuable in many real-world applications, but learning a generalizable model without overfitting to the few labeled datapoints is challenging. In this work, we focus on **F**ew-shot **L**earning with **A**uxiliary **D**ata (FLAD), a training paradigm that assumes access to auxiliary data during few-shot learning in hopes of improving generalization. Previous works have proposed automated methods for mixing auxiliary and target data, but these methods typically scale linearly (or worse) with the number of auxiliary datasets, limiting their practicality. In this work we relate FLAD to the explore-exploit dilemma that is central to the multi-armed bandit setting and derive algorithms whose computational complexity is independent of the number of auxiliary datasets, allowing us to scale to $100\times$ more auxiliary datasets than prior methods. We propose two algorithms – EXP3-FLAD and UCB1-FLAD – and compare them with prior FLAD methods that either explore *or* exploit, finding that the combination of exploration *and* exploitation is crucial. Through extensive experimentation we find that our methods outperform all pre-existing FLAD methods by 4% and lead to the first 3 billion parameter language models that outperform the 175 billion parameter GPT-3. Overall, our work suggests that the discovery of better, more efficient mixing strategies for FLAD may provide a viable path towards substantially improving generalization in few-shot learning. All of our code is available at github.com/alon-albalak/FLAD.

## 1 Introduction

Few-shot learning is an attractive learning setting for many reasons: it promises efficiency in cost and time, and in some scenarios data is simply not available due to privacy concerns or the nature of the problem. However, few-shot learning is also a challenging setting that requires a delicate balance between learning the structure of the feature and label spaces while preventing overfitting to the limited training samples [1, 2, 3]. One approach to improving the generalizability of models in the few-shot setting is **F**ew-shot **L**earning with **A**uxiliary **D**ata (FLAD), where additional auxiliary datasets are used to improve generalization on the target few-shot task [4, 5, 6, 7].

However, FLAD methods introduce their own challenges, including increased algorithmic and computational complexity. Specifically, incorporating auxiliary data during training introduces a large space of design choices (e.g. how and when to train on auxiliary data). Manually designing the curriculum for training on large quantities of auxiliary data is not feasible due to the combinatorially large search space, and hand-picking which auxiliary data to use based on heuristics (e.g. from the same domain or task as the target few-shot dataset) can lead to sub-optimal results [8]. Delegating such choices to an algorithm can lead to better solutions, as demonstrated in the transfer learning [8, 9, 10], meta-learning [11, 12], multi-task learning [13, 14, 15, 16], and auxiliary learning literature [4, 17]. However, prior auxiliary learning algorithms often assume that only 1-3 related auxiliary datasets are

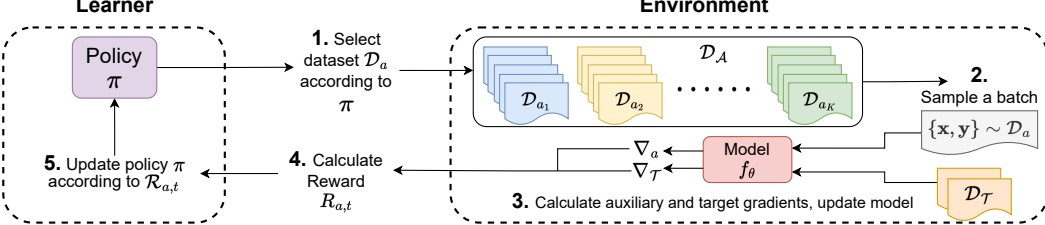

Figure 1: **Overview of few-shot learning with auxiliary data (FLAD) as a multi-armed bandit problem.** On the left is the learner which defines a policy $\pi$ that determines which auxiliary dataset to sample from. On the right is the environment that includes the set of auxiliary datasets $\mathcal{D}_\mathcal{A}$, target dataset $\mathcal{D}_\mathcal{T}$, and the model $f_\theta$. At each turn $t$, the following five steps take place, further described in Section 3.1: **1.** The learner selects an auxiliary dataset $\mathcal{D}_a$ according to its policy $\pi$. **2.** The environment samples a batch $\{\mathbf{x}, \mathbf{y}\} \sim \mathcal{D}_a$. **3.** The model $f_\theta$ calculates gradients for the sampled batch ($\nabla_a$) and the target dataset ($\nabla_\mathcal{T}$), then updates the parameters $\theta$. **4.** A reward $\mathcal{R}_{a,t}$ is calculated based on $\nabla_a$ and $\nabla_\mathcal{T}$. **5.** The learner updates $\pi$ based on $\mathcal{R}_{a,t}$.

available and design algorithms whose computational complexity grows linearly (or worse) with the number of auxiliary datasets [18, 8], motivating the search for more efficient methods as the number of auxiliary datasets grows.

To overcome the challenges of prior works, we desire a FLAD algorithm that **(1)** makes no assumptions on available auxiliary data a-priori (in-domain, on-task, quality, quantity, etc.), **(2)** scales well with the number of auxiliary datasets, and **(3)** adds minimal memory and computational overhead. We design algorithms that satisfy our desiderata by drawing inspiration from the central problem in multi-armed bandit (MAB) settings: the exploration-exploitation trade-off [19, 20]. We relate the set of auxiliary datasets to the arms of a MAB and tailor the classic EXP3 [21] and UCB1 [22] algorithms to fit the FLAD framework by designing three efficient gradient-based reward signals. The combination of our MAB-based algorithms and efficient gradient-based rewards allows us to scale to $100\times$ more auxiliary datasets than previous methods. Figure 1 provides a basic illustration of how we formulate FLAD as a MAB problem.

To empirically validate our approaches, we focus on few-shot training of language models and utilize P3 [23], a readily available resource with hundreds of auxiliary language datasets. We evaluate our methods on the same held-out tasks as the T0 language model [16] and show that, when using the same collection of auxiliary datasets, our algorithms outperform a directly fine-tuned T0 by 5.6% (EXP3-FLAD) and 5.7% (UCB1-FLAD) absolute. Furthermore, incorporating all available datasets in P3 (i.e. not just those used to train T0) increases the improvement to 9.1% and 9.2%. Finally, we compare models trained with our methods against state-of-the-art few-shot methods, finding that our methods improve performance by >3%, even though one model utilizes a large collection of unlabeled target dataset samples. Furthermore, to the best of our knowledge, our methods lead to the first 3 billion parameter model that improves over 175B GPT-3 using few-shot in-context learning.

In summary, our main contributions are:

- We connect FLAD to the MAB setting and focus on the exploration-exploitation trade-off by designing two algorithms, EXP3-FLAD and UCB1-FLAD along with three reward functions that are both simple and efficient (in space and computational complexity).
- We empirically validate that our methods improve few-shot performance of pretrained language models and show that strategies that employ only exploration *or* exploitation lead to sub-optimal performance.
- We perform case studies to better understand the dynamics of our reward functions and their interaction with the dynamics of large language model training.

## 2   Related work

A long history of works have found success when combining auxiliary data with target data [4, 24, 6, 25, 26, 5, 18, 7, 27, 28, 8]. Some works have explored the addition of auxiliary learning objectives to aid the learning of the target task [24, 26, 25, 5, 17]. More similar to our work are

methods that perform auxiliary learning by introducing additional data sources beyond the target data [4, 6, 18, 7, 27, 28, 8]. As opposed to the few-shot setting on which this work focuses, previous works have studied auxiliary learning in settings with large quantities of target data. For example, Chen et al. [18] and Verboven et al. [7] assume access to 10,000 labeled target samples, Ivison et al. [28] and Lin et al. [27] assume access to 1,000s of unlabeled target samples, and Du et al. [6] and Albalak et al. [8] assume access to 100s of labeled target samples. Additionally, many of the previous works that study auxiliary learning have only considered settings with 1-3 auxiliary datasets [6, 18, 7, 8]. For example, Verboven et al. [7] propose a task-weighting method that requires solving a system of equations that becomes underspecified with multiple auxiliary tasks, limiting their method to only a single auxiliary task. Furthermore, Chen et al. [18] experiment with 3 auxiliary tasks because their method requires learning a target-aware classifier for each source task, so the computation scales as $O(|\mathcal{A}||\mathcal{T}|)$ where $|\mathcal{A}|$ is the number of auxiliary tasks and $|\mathcal{T}|$ is the number of target tasks, making it impractical to scale to large numbers of source and target tasks. In this work, we focus on improving auxiliary learning with very few target samples (20-70 samples) by scaling up the number of auxiliary datasets orders of magnitude greater than previous work. In order to scale up the learning process, efficiency is a central concern of this work, unlike prior works.

Data selection studies a similar (but distinct) problem where the goal is to selectively utilize a subset of a single large dataset rather than selecting data from auxiliary datasets. Recent research on data selection has found that intelligent data selection can provide significant improvements to model performance [29, 30, 31, 32].

## 3 Multi-armed bandits for few-shot learning with auxiliary data

In this section, we first define the few-shot learning with auxiliary data (**FLAD**) setting. Then, we formulate FLAD as a multi-armed bandits (**MAB**) problem, shown in Figure 1. Next, we define reward functions that are efficient to compute and appropriate for FLAD. Finally, we describe our adaptations of two popular MAB algorithms: EXP3-FLAD and UCB1-FLAD.

### 3.1 Setup

**FLAD problem setting.** Few-shot learning with auxiliary data (FLAD) fits into the following setting: assume access to a large set of auxiliary datasets $\mathcal{D}_{\mathcal{A}}$ where, for all $a \in \mathcal{A}$, $\mathcal{D}_a$ is an individual auxiliary dataset. Given a small quantity of data belonging to a target dataset $\mathcal{D}_{\mathcal{T}}$, the goal of FLAD is to find parameters $\theta$ of a model $f_\theta$ that achieve high performance on the unknown distribution underlying $\mathcal{D}_{\mathcal{T}}$ while utilizing only the available data, $\mathcal{D}_{\mathcal{T}} \cup \mathcal{D}_{\mathcal{A}}$.

**Formulating FLAD as MAB.** In this work, we adopt the multi-armed bandit (MAB) setting by formulating FLAD as a Markov decision process [33] and defining a learner and environment, illustrated in Figure 1. The learner consists of a policy $\pi$ defining a selection strategy over all $\mathcal{D}_a \in \mathcal{D}_{\mathcal{A}}$. The environment consists of the target dataset $\mathcal{D}_{\mathcal{T}}$, auxiliary datasets $\mathcal{D}_{\mathcal{A}}$, and model $f_\theta$. In this formulation the learner interacts with the environment over $N$ rounds. At each round $t$ the learner selects one of the environment's $|\mathcal{A}|$ datasets $\mathcal{D}_a \in \mathcal{D}_{\mathcal{A}}$. Next, the environment samples a batch $\{\mathbf{x}, \mathbf{y}\} \sim \mathcal{D}_a$ and calculates the gradient w.r.t. $\theta$ using task-appropriate loss function as $\nabla_a = \nabla_\theta \mathcal{L}(f_\theta, \mathbf{x}, \mathbf{y})$. Then, the environment computes the target gradient $\nabla_{\mathcal{T}} = \nabla_\theta \mathcal{L}(f_\theta, \mathcal{D}_{\mathcal{T}})$, and updates model parameters w.r.t. $\nabla_{\mathcal{T}} + \nabla_a$. Finally, the learner uses a gradient-based reward $\mathcal{R}_{a,t}(\nabla_a, \nabla_{\mathcal{T}})$ to update its policy $\pi$. See Appendix A and Lattimore & Szepesvári [34] for further details on multi-armed bandits.

**Designing the reward functions.** We design the reward function $\mathcal{R}$ with our desiderata in mind. To ensure that our algorithm adds minimal memory and computational overhead we consider rewards that utilize information intrinsic to the model and the losses being optimized, not an external model or metric (e.g. accuracy or BLEU). In this work we propose three gradient-based reward functions inspired by previous works: **gradient alignment** [6, 24, 35], **gradient magnitude similarity** [36, 37], and their aggregation. Formally, at turn $t$ let $\nabla_a$ be the gradient of the auxiliary batch and $\nabla_{\mathcal{T}}$ be the target dataset gradient. **Gradient alignment** is defined as $\mathcal{R}_{a,t}^{GA} = \frac{\nabla_a \cdot \nabla_{\mathcal{T}}}{\|\nabla_a\|_2 \|\nabla_{\mathcal{T}}\|_2}$, i.e. the cosine similarity between the gradients of the sampled auxiliary dataset batch and the whole target dataset. **Gradient magnitude similarity** is defined as $\mathcal{R}_{a,t}^{GMS} = \frac{2\|\nabla_a\|_2 \|\nabla_{\mathcal{T}}\|_2}{\|\nabla_a\|_2^2 + \|\nabla_{\mathcal{T}}\|_2^2}$ so that when the two gradients have equal magnitude, this value is equal to 1 and as the magnitudes differ the value goes to zero. In

addition to the individual reward functions, we also consider an aggregate reward. To ensure that the aggregate is not dominated by either individual reward, we normalize $\mathcal{R}^{GA} \in [0, 1]$, the same range as $\mathcal{R}^{GMS}$ and define the aggregate to be their sum: $\mathcal{R}_{a,t}^{AGG} = \frac{1+\mathcal{R}_{a,t}^{GA}}{2} + \mathcal{R}_{a,t}^{GMS}$. We provide further discussion on the design of reward functions in Section 6.

## 3.2   Adapting the EXP3 algorithm.

**EXP3 Background**   We base our first algorithm, EXP3-FLAD, on the EXP3 algorithm [21] ("*Exp*onential-weight algorithm for *Expl*oration and *Expl*oitation"). EXP3 targets the adversarial MAB setting, which assumes that the reward-generating process is controlled by an adversary who is given access to the learner's policy $\pi$ and determines the sequence of rewards, $(R_{a,t})_{t=1}^{N}$, for each arm prior to play [38]. We consider the adversarial MAB formulation due to the highly non-convex loss landscape of deep neural networks and our use of stochastic gradient descent-based optimization methods. These factors imply that we cannot guarantee our rewards to be stationary, independent, or follow any particular distribution (e.g. Gaussian). Further details on adversarial MAB are included in Appendix A and in [21].

In EXP3-FLAD, the learner selects arms according to a Gibbs distribution based on the empirically determined importance-weighted rewards of arms [39]. To allow for exploration, we mix the Gibbs distribution with a uniform distribution [21]. Formally, let $\mathcal{E}_t$ be the exploration rate at turn $t$ and, recalling that $K = |\mathcal{A}|$ is the number of auxiliary datasets, then $\pi$ defines the probability of selecting a given arm $a \in \mathcal{A}$ as the linear combination of Gibbs and uniform distributions $\pi_t(a) = (1 - K\mathcal{E}_t)\frac{\exp(\mathcal{E}_{t-1}\hat{R}_a)}{\sum_{a'}\exp(\mathcal{E}_{t-1}\hat{R}_{a'})} + \mathcal{E}_t$ where $\hat{R}_{a,t}$ is the importance weighted reward $\hat{R}_{a,t} = \hat{R}_{a,t-1} + \frac{R_{a,t}}{\pi_{t-1}(a)}$. We want the learner to explore more in early training than in later stages, so we use a decaying exploration rate $\mathcal{E}_t = \min\left\{\frac{1}{K}, \sqrt{\frac{\ln K}{K \cdot t}}\right\}$ as proposed by Seldin et al. [39]. The use of an importance-weighted estimated reward compensates the rewards of actions that are less likely to be chosen, guaranteeing that the expected estimated reward is equal to the actual reward for each action. EXP3-FLAD is designed to be nearly optimal in the worst case, but due to the exploration rate it will select "bad" actions at a rate of $\mathcal{E}_t$. The exploration of EXP3-FLAD combined with importance-weighting allows the policy to handle non-stationary reward-generating processes.

## 3.3   Adapting the UCB1 algorithm.

**UCB1 background.**   While EXP3-FLAD is applicable in unconstrained settings with highly stochastic and non-stationary rewards, it can be outperformed by other algorithms in settings that *are* constrained. One such algorithm is the upper confidence bound (UCB1) algorithm [22], which was originally designed to be optimal for stationary, normally distributed reward functions. Nevertheless, variants of UCB1 have been demonstrated to be effective in a range of settings, such as those involving non-stationary, sub-Gaussian, or heavy-tailed distributions [40, 41]. The UCB1 algorithm and its variants assign each arm a value called the upper confidence bound based on Hoeffding's inequality [42] and are based on the principle of *optimism in the face of uncertainty*, meaning that with high probability the upper confidence bound assigned to each arm is an overestimate of the unknown mean reward.

In UCB1-FLAD, the learner greedily selects arms according to their upper confidence bound. UCB1 is originally designed for stationary reward-generating processes, so to accommodate non-stationarity we include an exponential moving average when estimating the mean reward for a given arm. Formally, let $R_{a,t}$ be the observed reward for arm $a$ at turn $t$, then we calculate the estimated mean reward as $\hat{R}_a = (1 - \beta)\hat{R}_a + \beta R_{a,t}$ where $\beta$ is the smoothing factor. Then, we define the upper confidence bound to be $UCB_{a,t} = \hat{R}_a + \sqrt{\frac{2\ln t}{n_a}}$. In the original MAB setting all interactions with the environment occur online, but FLAD is a unique situation where the learner can interact with the auxiliary data prior to training. To take advantage of this, rather than initializing estimated rewards with a single mini-batch, we initialize them with larger data quantities to improve the approximation of the true dataset gradients. This is done for each auxiliary dataset by calculating the gradient $\nabla_a = \nabla_\theta \mathcal{L}(f_\theta, \mathbf{x}, \mathbf{y})$, where the number of samples in $\{\mathbf{x}, \mathbf{y}\}$ can be significantly larger than a mini-batch, and can be up to the size of the full dataset. In practice, we use 1,000 examples which is computed in $\sim 2$ minutes on a single GPU.

**Algorithms** The EXP3-FLAD and UCB1-FLAD algorithms are visualized in Figure 1. At each turn, both methods will first select an auxiliary dataset $\mathcal{D}_a$. EXP3-FLAD first computes the current exploration rate $\mathcal{E}_t$ and samples $\mathcal{D}_a$ according to the distribution defined by $\pi_t(\mathcal{A})$, while UCB1-FLAD greedily selects $\mathcal{D}_{a^*}$ corresponding to the arm with largest upper confidence bound, $a^* = \arg\max_{a \in \mathcal{A}} UCB_{a,t}$. Next, for both methods, the environment samples a batch from the selected dataset, $\{\mathbf{x}, \mathbf{y}\} \sim \mathcal{D}_a$, and calculates the gradient $\nabla_a = \nabla_\theta \mathcal{L}(f_\theta, \mathbf{x}, \mathbf{y})$. Let $G$ be the number of rounds between model updates, then the previous steps will repeat $G$ times, at which point the environment calculates the gradient of the target dataset $\nabla_\theta \mathcal{L}(f_\theta, \mathcal{D}_\mathcal{T})$ and updates the model w.r.t. $\nabla_\mathcal{T} + \sum_a \nabla_a$. Finally, EXP3-FLAD calculates the importance-weighted reward for each auxiliary batch using the observed rewards, while UCB1-FLAD calculates the smoothed estimated mean reward. Pseudocode is found in Appendix B.

## 4 Experimental setup

**Models.** For our experiments, we utilize encoder-decoder Transformer models from the T5 family of pre-trained language models [43]. Specifically, we experiment with LM-adapted T5 (T5-LM) and T0. The T5-LM model further trains the T5.1.1 model for 100,000 steps (corresponding to 100B tokens) from the C4 dataset [43] on the prefix language modeling objective [44]. The T0 model was initialized from T5-LM and further trained on a multitask mixture of prompted datasets as described by Sanh et al. [16]. We repeat each experiment with T5-LM XL (hereafter **T5-XL**) and **T0-3B** as our base model. Both models use the same architecture with 2.85 billion parameters, and we used model checkpoints from Hugging Face Transformers [45]).

**Target datasets.** We obtain all datasets from Hugging Face Datasets[1], and cast them to the text-to-text format by applying prompt templates from the Public Pool of Prompts (P3) [23] that was used to train T0. To evaluate our few-shot methods, we utilize the same held-out datasets as T0, which cover four distinct tasks: **sentence completion** (COPA [46], HellaSwag [47], Story Cloze [48]), **natural language inference** (ANLI [49], CB [50], RTE [51]), **coreference resolution** (WSC [52], Winogrande [53]), and **word sense disambiguation** (WiC [54]). For each dataset, we randomly sample five few-shot splits from their training data, containing the same number of training examples as previous works, between 20 to 70 [55, 56]. We further divide each split into equal training and validation partitions for true few-shot learning [57](e.g. 10 train and 10 validation samples for HellaSwag). Only ANLI datasets have a publicly available test set, so for all other datasets we evaluate models on the original validation set (not utilized for few-shot training or validation).

**Auxiliary datasets.** We compare the performance of our methods using two sets of auxiliary data and never include any of the target datasets as part of auxiliary data. First, we use the collection of datasets used for multitask training of T0 (henceforth referred to as T0Mix), including 35 unique datasets covering question answering, sentiment analysis, topic classification, summarization, paraphrase detection and structure-to-text. Second, we utilize all datasets in P3 [23] (which forms a superset of T0Mix) and prevent data leakage by filtering out datasets that overlap with any target dataset, leading to 260 available datasets (list in Appendix H). For each auxiliary dataset, we use at most 10,000 of the dataset's examples.

**Baseline methods.** We compare our proposed methods with several FLAD and non-FLAD baselines. **Target-Only** (non-FLAD) directly fine-tunes the base model on the target dataset (i.e. without using auxiliary data). **Explore-Only** [8] is a commonly used FLAD method which simultaneously trains on auxiliary and target data by mixing auxiliary datasets equally. Originally called Multitask in [8], we call this Explore-Only because it is equivalent to continuously exploring auxiliary data and never exploiting knowledge of its relation to the target data. **Exploit-Only** computes gradient alignment prior to training (as in UCB1), and multitask-trains the model by mixing auxiliary datasets according to a Gibbs distribution over the alignments (similar to that in EXP3), resulting in an algorithm that exploits the relations determined prior to training, but never exploring. Both explore- and exploit-only mix target and auxiliary data with a ratio of $M$ times the highest auxiliary sampling probability. For instance, explore-only with $M = 5$ and $\mathcal{D}_\mathcal{A} = $ T0Mix has a $1/35$ probability to sample auxiliary dataset $\mathcal{D}_a \in \mathcal{D}_\mathcal{A}$ and a $5/35$ probability for the target dataset. **Loss-Scaling** [6] is a FLAD method similar to EXP3 and UCB1; the main difference being that it scales auxiliary batch losses by their gradient alignment instead of modifying sampling probabilities. Du et al. [6] originally propose to

---

[1]https://huggingface.co/datasets

| Training Method | BASE MODEL \ Auxiliary Data | T5-XL T0Mix | T5-XL P3 | T0-3B T0Mix | T0-3B P3 |
|---|---|---|---|---|---|
| Target-Only | | 52.82 | | 56.44 | |
| Loss-Scaling [6] ($GA$) | | 53.22 | 55.19 | 59.47 | 60.66 |
| Loss-Scaling [6] ($GMS$) | | 55.98 | 56.40 | 60.47 | 60.70 |
| Explore-Only [8] | | 59.18 | 60.64 | 61.17 | 62.77 |
| Exploit-Only [8] | | 59.79 | 60.49 | 60.87 | 62.87 |
| EXP3-FLAD ($\mathcal{R}^{GA}$) | | 61.50 | 64.07 | 62.87 | 65.98 |
| UCB1-FLAD ($\mathcal{R}^{GA}$) | | 62.01 | 65.52 | 62.35 | 66.29 |
| EXP3-FLAD ($\mathcal{R}^{GMS}$) | | 61.72 | 65.57 | 62.78 | 65.51 |
| UCB1-FLAD ($\mathcal{R}^{GMS}$) | | 61.67 | 65.21 | 62.85 | 66.00 |
| EXP3-FLAD ($\mathcal{R}^{AGG}$) | | 62.05 | 65.47 | 62.84 | **66.84** |
| UCB1-FLAD ($\mathcal{R}^{AGG}$) | | **62.08** | **65.63** | **62.93** | 66.29 |

Table 1: **Main results.** Each cell contains the score of training a base model (top row) with auxiliary data (second row) using the specified training method (left column), averaged across 11 target datasets on 5 random seeds (each cell is the average of 55 experiments). Target-Only does not utilize auxiliary data. **Bolded** scores are those with highest mean for a given base model and auxiliary dataset (column-wise), underlined scores are those where a Wilcoxon rank-sum test fails to find significant difference from the highest score ($p > 0.05$). Expanded results are found in Appendix D.

use gradient alignment (**Loss-Scaling** ($GA$)), but we also propose a version that scales losses by gradient magnitude similarity (**Loss-Scaling** ($GMS$)).

**Training details.**  For the target-only baseline, we use learning rates in $\{1e\text{-}4, 3e\text{-}4\}$. For all other methods, we always use a learning rate of $1e\text{-}4$. For target-, explore-, and exploit-only baselines we use batch sizes in $\{32, 128\}$. For loss-scaling, EXP3-FLAD, and UCB1-FLAD we use mini-batches of 8 samples and let $G$ be in $\{4, 16\}$ to match the batch size of all methods. For explore- and exploit-only, we use a target dataset mixing ratio of $M \in \{1, 5, 10\}$. For all experiments we use the Adafactor optimizer [58] and validation-based early stopping for model checkpoint selection. In preliminary experiments we consider rewards using gradients from various model partitions: the full model, encoder-only, decoder-only, and the weights of the output vocabulary matrix (language modeling head). We find that using the parameters from the language modeling head provides the best performance and contains only 2.3% of the full model parameters, significantly reducing memory consumption. For UCB1-FLAD we found the smoothing factor $\beta = 0.9$ to work well in preliminary experiments and initialize auxiliary dataset gradient alignment using 1,000 samples from each auxiliary dataset. Additional implementation details can be found in Appendix C

**Experiment procedure.**  The FLAD experiment process involves training a model that is specialized for each target dataset. For each proposed method and baseline, we train and evaluate a model on each of the 11 target datasets. We repeat training and evaluation on 5 random seeds and include the aggregated results in Table 1. Each cell shows the accuracy averaged across all 55 (11 target datasets, 5 random seeds) experiments. This experimental process is performed for each training method on both models and auxiliary datasets. We include the non-aggregated results in Appendix D.

## 5 Findings and analysis

In Table 1 we compare the empirical results of our MAB-based methods (EXP3-FLAD and UCB1-FLAD) and corresponding baselines on 11 target datasets (expanded results in Appendix D. For each base model and auxiliary data combination (each column) EXP3-FLAD and UCB1-FLAD outperform all the baselines. In fact, we find that *for every single task* our methods always perform equal to or better than the baselines. This demonstrates that our MAB-based methods provide a strong improvement in few-shot generalization over previous FLAD methods. For a fair comparison where each method utilizes equal data, we compare the performance of Target-Only using T0 and T0Mix (56.44) against the proposed FLAD methods and baselines using T5 and T0Mix (left column). From this comparison it becomes clear that Loss-Scaling actually does worse than multitask training followed by direct fine-tuning by 0.5-3.2%. However, we do find that the remaining FLAD methods lead to improvements (between 2.7-5.6% absolute improvement). We find small performance

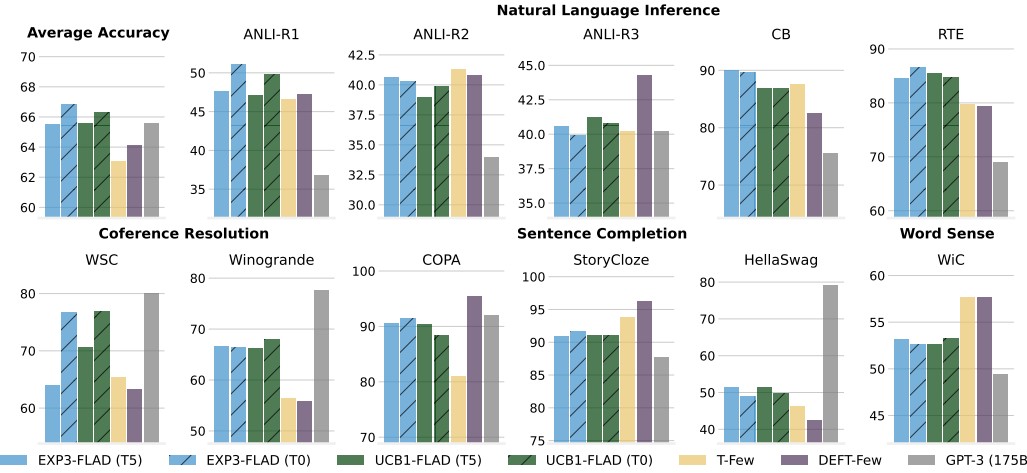

Figure 2: **Comparison of state-of-the-art few-shot methods with FLAD methods trained on P3 using** $\mathcal{R}^{\mathbf{AGG}}$. T-Few scores are from [56]. DEFT-Few scores are from [28]. GPT-3 scores are from [55] and utilize few-shot in-context learning. All models utilize the same number of few-shot examples and (other than GPT-3) have 3B parameters.

differences between EXP3-FLAD and UCB1-FLAD across the three reward functions. In general, $\mathcal{R}^{AGG}$ leads to the best performance, but we perform a two-sided Wilcoxon rank-sum test to check for significance between average scores and find that the other rewards frequently have no significant difference ($p > 0.05$).

**The importance of prioritized sampling.** Loss-Scaling was originally proposed for use with only a single auxiliary dataset and it was unclear, a priori, how it would cope with larger quantities. Additionally, Du et al. [6] purposefully choose an auxiliary dataset that is related to the target, while in our setting we make no such assumptions. We find that our methods outperform Loss-Scaling methods by 6.3% on average. In Figure 3 (and Figure 5 in Appendix E) we show that, over the course of training, the value of gradient alignments and gradient magnitude similarities for most datasets will converge to 0, leading to very small gradient updates for Loss-Scaling. More importantly, *the auxiliary data that is relevant to the target task is seen less frequently for Loss-Scaling* than our MAB-based methods. This can be seen by comparing the difference in performance of Loss-Scaling methods when using less (T0Mix) vs. more (P3) auxiliary data. We find that, at best, Loss-Scaling ($GA$) improves 2% when using T5 and, at worst, only 0.2% for Loss-Scaling ($GMS$) with T0. This is compared with the notable improvements of EXP3-FLAD and UCB1-FLAD of 2.6-4% when considering the same data increase from T0Mix to P3.

**The importance of exploration *and* exploitation.** Interestingly, we expected that Exploit-Only would outperform the Explore-Only method because it utilizes relational information between the target and auxiliary tasks, but find no statistical difference between the methods (two-sided Wilcoxon rank-sum test gives $p > 0.05$). Furthermore, when comparing the ability to leverage additional auxiliary data (i.e. going from T0Mix to all of P3), we find that the improvement for Explore- and Exploit-Only methods is minimal with only 0.7-2% improvement. On the other hand, EXP3-FLAD and UCB1-FLAD show a notable improvement of 2.6-4%, emphasizing the importance of both exploration *and* exploitation, particularly when dealing with large collections of auxiliary data.

**FLAD provides improved generalization over non-FLAD methods.** Next, we compare the performance of our best models trained on P3 using $\mathcal{R}^{AGG}$ with state-of-the-art few-shot methods: T-Few, DEFT-Few, and GPT-3. T-Few [56] is a variant of the T0-3B model that multi-task pre-trains parameter-efficient $(IA)^3$ modules followed by target-only fine-tuning of the $(IA)^3$ modules. DEFT-Few [28] is a variant of the T5-XL model that uses retrieved auxiliary data for multi-task training. It first trains a T5-XL model on the 500 nearest neighbor samples from P3 using 1000 unlabeled target dataset samples, and then performs few-shot target-only fine-tuning with the $(IA)^3$ modules from Liu et al. [56]. Finally, we also compare against the 175 billion parameter variant of

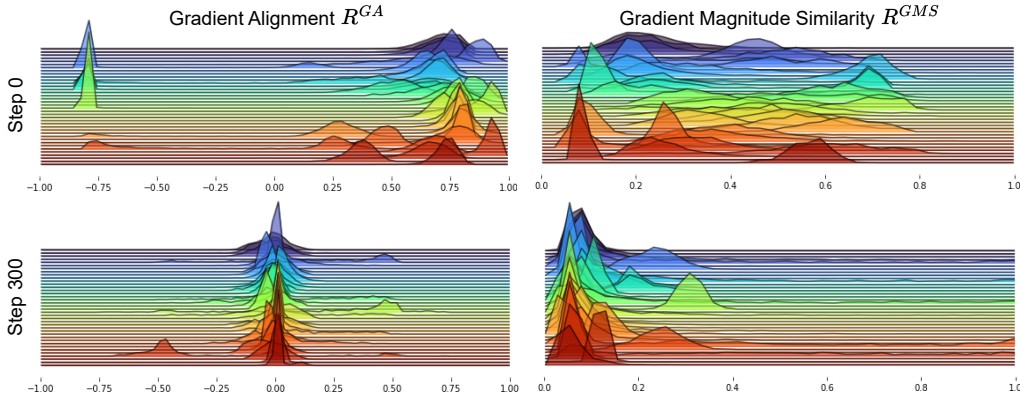

Figure 3: **Reward distributions** of $R^{GA}$ and $R^{GMS}$ prior to training (step 0) and after 300 gradient updates for the T5-XL model with T0Mix as the auxiliary dataset and WSC [52] as the target dataset. Each quadrant shows the histograms of reward distributions for all 35 auxiliary datasets. By step 300 most auxiliary datasets provide 0 reward, while only the few remaining "beneficial" datasets provide positive rewards. Results from every 100 gradient updates are shown in Figure 5 in Appendix E.

GPT-3 [55], which utilizes in-context learning. We find that, on average, models trained using our FLAD-based methods outperform all other methods and, to the best of our knowledge, our methods lead to the first 3 billion parameter model that outperforms GPT-3 on this dataset mixture (previous smallest models have 11 billion parameters), despite using $62.5$ times fewer parameters than GPT-3. Additionally, we find that our FLAD-based methods provide robust performance across datasets, achieving the best or second-best performance on $8/11$ datasets, and never performing worst. The use of task-specific modules lead T-Few and DEFT-Few to significant improvements over target-only fine-tuning, preventing the models from ending up in poor local minima. However, these results demonstrate that with the same data, simultaneously fine-tuning with auxiliary and target data leads to improved few-shot generalization, providing a complementary means of improving performance.

**Investigating the Reward-Generating Processes.** In Section 3.2, we mention that due to the highly non-convex loss landscape and the use of stochastic gradient descent-based optimization techniques, we cannot ensure that our reward generating process is stationary, independent across auxiliary datasets, or follows a normal distribution. To gain a deeper understanding of our reward-generating processes, we examine the distribution of each reward using 5,000 samples from all 35 auxiliary datasets of T0Mix and 32 samples from a few-shot target dataset, WSC [52]. The resulting histograms at every 100 steps can be found in Appendix E, and Figure 3 shows an abbreviated version. The left side of Figure 3 demonstrates that for $\mathcal{R}^{GA}$, almost every dataset yields a Gaussian reward distribution, with a few multi-modal distributions. Notably, WikiBio [59] (dark orange) exhibits peaks at 0.25 and -0.75. Interestingly, $\mathcal{R}^{GA}$ results in polarized rewards across datasets, with minimal distribution density between -0.75 and 0.25. In contrast, the right side of Figure 3 displays more non-Gaussian distributions for $\mathcal{R}^{GMS}$, as well as flatter distributions compared to $\mathcal{R}^{GA}$. Remarkably, we observe that $\mathcal{R}^{GA}$ produces more stationary reward distributions, as the distribution for almost every dataset (30/35) converges rapidly towards 0 after only 100 steps. Although most distributions for $\mathcal{R}^{GMS}$ also converge towards 0, the convergence occurs at a slower pace, taking nearly 500 steps.

**Probing the training dynamics.** To better understand the training dynamics of our proposed methods, we perform a case study on T5-XL with T0Mix and $\mathcal{R}^{GA}$ and find two datasets where either algorithm improves significantly over the other (full details and figures in Appendix F). First, we study RTE, where UCB1-FLAD outperforms EXP3-FLAD. We calculate the empirical distribution of samples seen from each auxiliary dataset and find that EXP3-FLAD samples nearly uniformly from all datasets while UCB1-FLAD forms a bimodal sampling distribution with peaks at 2.5% and 3.25% (30% relative difference). The uniformity of the EXP3-FLAD distribution is counterintuitive, as we do find that it achieves separation between auxiliary tasks in the cumulative estimated reward (as shown in Figure 7), but this does not lead to separation in the sampling probability space. Additionally we find that even on COPA, where EXP3-FLAD outperforms UCB1-FLAD, EXP3-FLAD still achieves

good separation between cumulative estimated rewards, but has a unimodal sampling distribution, while UCB1-FLAD does not have as clear of a bimodal distribution as in RTE. The difference in empirical sampling distributions is likely due to the difference between the greedy policy of UCB1-FLAD and the stochastic policy of EXP3-FLAD. Empirically, we find that EXP3-FLAD very rarely assigns an auxiliary dataset a probability $< 1\%$, leading to many "bad" batches over the course of thousands of turns. On the other hand, the optimistic policy of UCB1-FLAD spends much less time exploring and will sample "bad" batches much less frequently.

**The effect of scaling** $|\mathcal{A}|$  To assess the scalability of our proposed methods, we conduct a study by varying the size of $|\mathcal{A}| \in \{35, 75, 125, 175, 225, 260\}$. For each value of $|\mathcal{A}|$, we consistently include the 35 datasets from T0Mix and randomly select additional auxiliary datasets from P3 until we reach the desired $|\mathcal{A}|$. The study is performed on the same 11 target datasets as the main study, using the T0 base model and $\mathcal{R}^{AGG}$ reward. The experiment is repeated with three random seeds. Figure 4 shows the mean across the 11 target datasets, along with the standard deviation between seeds.

We find that both EXP3-FLAD and UCB1-FLAD experience a sharp increase from $|\mathcal{A}| = 35$ to 75. In addition, we observe improvements up to the maximum value of $|\mathcal{A}| = 260$, ultimately improving accuracy by 2.54 for EXP3-FLAD and 3.12 for UCB1-FLAD when transitioning from 35 to 75 datasets, with further increases of 1.54 for EXP3-FLAD and 0.47 for UCB1-FLAD when increasing $|\mathcal{A}|$ from 75 to 260.

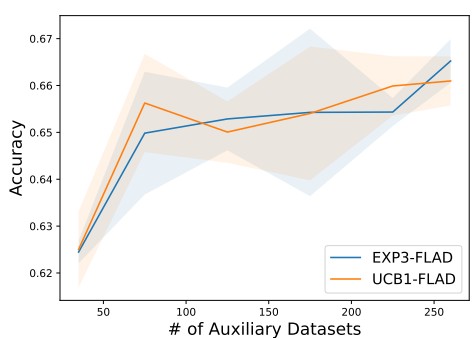

Figure 4: **The effect of scaling** $|\mathcal{A}|$ **on target task performance**. Each line represents mean score across 11 datasets and three random seeds, with shaded regions falling between one standard deviation of the mean.

# 6  Discussion

**Discussion on reward functions.**  In FLAD we want to prioritize training on auxiliary datasets with similar solution spaces as the target task without overfitting to the few-shot target data, and our reward functions are designed to serve this goal. To better understand the reward signal of our aggregate reward, $\mathcal{R}^{AGG}$, we examine four combinations of rewards: low $\mathcal{R}^{GA}$ and $\mathcal{R}^{GMS}$, high $\mathcal{R}^{GA}$ but low $\mathcal{R}^{GMS}$, low $\mathcal{R}^{GA}$ but high $\mathcal{R}^{GMS}$, and high $\mathcal{R}^{GA}$ and $\mathcal{R}^{GMS}$. When both rewards are high, we can assume that the auxiliary gradient is useful. However, when one reward is high and the other is low, it is difficult to draw conclusions as a high $\mathcal{R}^{GA}$ on its own means the auxiliary gradient will update weights in the right direction, but low $\mathcal{R}^{GMS}$ can mean that we significantly overshoot *or* undershoot the target, where overshooting can be much more detrimental than undershooting. If both $\mathcal{R}^{GA}$ and $\mathcal{R}^{GMS}$ are small, we know the auxiliary gradient leads us away from the target solution space, but we don't know if its magnitude is much larger or smaller than the target. At the beginning of training, we can't know if the target or auxiliary gradient has larger magnitude, but as training progresses, it becomes significantly more likely that the auxiliary gradient is greater than the target. Thus, when both $\mathcal{R}^{GA}$ and $\mathcal{R}^{GMS}$ are low, we are likely to be pulled far from our current solution.

This work uses training set-based rewards, but validation set-based rewards are also possible. One downside of validation-based rewards is they calculate validation score frequently, which increases computational complexity. Additionally, we focus on the few-shot setting and use validation-based early stopping. If we use a validation-based reward, then to prevent overfitting we will need to further split the data into 3 partitions: train, early-stopping validation, and reward-validation.

**Choice of baselines.**  With respect to the number of auxiliary datasets $|\mathcal{A}|$ and target datasets $|\mathcal{T}|$, our methods and the baselines we compare against have a computational complexity of $O(|\mathcal{T}|)$, independent of $|\mathcal{A}|$. For our model and these baselines, the models we train require $\sim 6$ GPU-hours per target dataset. If we were to consider a baseline whose computation grows linearly w.r.t. $|\mathcal{A}|$,

$O(|\mathcal{A}||\mathcal{T}|)$ (e.g. [18]), these experiments would not be feasible without a large amount of hardware: *Training such a model with T0Mix would take over 200 GPU-hours (over 8 GPU-days) for a single target dataset*, and over 1500 GPU-hours (*over 2 GPU-months*) when using all of P3.

**Why we don't include theoretical guarantees.** The design of MAB algorithms generally comes with theoretical proofs of regret bounds, but in this work we omit such analysis. Although we *could* make guarantees on the regret bounds of our algorithms, they would not be meaningful because they would be with respect to the rewards, not the accuracy on a held-out dataset (which is the quantity we actually care about).

**How does FLAD relate to few-shot learning and multitask learning?** Both few-shot learning and FLAD are concerned with optimizing model performance on a single target task with a limited number of examples from the target task. In few-shot learning, the model is given only the target task data $\mathcal{D}_{\mathcal{T}}$ and there is no auxiliary data. Effectively, $\mathcal{D}_{\mathcal{A}}$ is the empty set for few-shot learning. In contrast, for the FLAD setting $|\mathcal{D}_{\mathcal{A}}| > 1$. Based on the findings from this study, we highly recommend that practitioners utilize auxiliary data when it is available.

Multitask learning is concerned with optimizing a model for performance on multiple target datasets simultaneously. This is in direct opposition with the FLAD methods presented here, which aim to optimize a model for a single target task. However, it is possible to extend our MAB-based methods to optimize for multiple target tasks simultaneously by aggregating multiple rewards. We believe this would make for an interesting future study.

**Limitations.** One of the implicit assumptions in the FLAD setting (made by this work and all prior works) is that there is at least *some* auxiliary data that will be useful for the target task. However, one of the main distinctions of our methods from prior works in the FLAD setting is that prior works make a strong assumption that all auxiliary data are useful, and thus appropriate auxiliary datasets must be hand-picked by humans. On the other hand, our methods allow for only a small portion of the auxiliary data to be useful – our proposed algorithm explores to find useful auxiliary datasets and then exploits them.

# 7   Conclusion

Recall the desiderata for our algorithm, expressed in the introduction: our algorithm should **(1)** make no assumptions on the available auxiliary data a-priori, **(2)** scale well with the number of auxiliary datasets, and **(3)** add minimal memory and computational overhead. **(1)** When designing our algorithm, we purposefully formulate the problem as a multi-armed bandit. MAB algorithms, in general, make no assumptions on the quality of rewards and, in particular, EXP3 even assumes that the auxiliary datasets will play an adversarial role when returning rewards. **(2)** As previously mentioned, our algorithms have a single-turn computational complexity that is independent of the number of auxiliary datasets. **(3)** Finally, our method adds minimal computational overhead beyond usual training computations. Every gradient that we utilize for our reward functions are also used to update the model, adding no additional computations. The only computational overhead is to compute gradient alignment (three vector dot products, two scalar square roots, and two scalar multiplications) or magnitude similarity (four vector dot products, two scalar square roots, three scalar multiplications, and one scalar addition). Additionally, our method adds a small amount of memory overhead, used to store gradients between model updates. Our rewards consider only the gradient w.r.t the language modelling head and, in practice, require 0.25Gb per auxiliary gradient to store, slightly increasing the space complexity above standard fine-tuning.

The methods proposed in this work demonstrate the effectiveness of simultaneous training on auxiliary and target datasets in few-shot settings, continuously updating beliefs by exploring *and* exploiting auxiliary data, and framing FLAD as a MAB problem. We further showed that by satisfying our desiderata, we are able to scale up FLAD to hundreds of auxiliary datasets and outperform traditional few-shot fine-tuning and in-context learning methods. While the presented algorithms satisfy our desiderata, the findings from this study can inform future work to further improve upon these methods in a number of ways, such as improving the reward function and reducing the space complexity.

## Acknowledgements

This material is based on work that is partially funded by an unrestricted gift from Google. This work was supported by the National Science Foundation award #2048122. The views expressed are those of the authors and do not reflect the official policy or position of the US government. Additionally, we thank the Robert N. Noyce Trust for their generous gift to the University of California via the Noyce Initiative.

We would also like to thank Dheeraj Baby for insightful advising on multi-armed bandit settings as well as Lucio Dery and Jiachen Li for invaluable feedback on early drafts of this work.

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

# A  Multi-armed bandits

The Multi-Armed Bandit (**MAB**) setting is a problem from machine learning where a learner interacts with an environment over $N$ rounds by following a policy $\pi$. At each round $t$ the learner chooses one of the environment's $K$ arms, $a \in \mathcal{A}$ where $K = |\mathcal{A}|$, after which the environment provides a reward $R_t$. Rewards for unplayed arms are not observed. The goal of the learner is to adopt a policy $\pi$ that selects actions that lead to the largest cumulative reward over $N$ rounds, $R = \sum_{t=1}^{N} R_t$. In this work we assume a finite $K$ and that the underlying reward distribution of each arm may have a variety of properties (e.g. stochasticity or stationarity) depending on the exact scenario, leading to different optimal policies [34].

**Adversarial MAB.** The adversarial MAB setting assumes that the reward-generating process is controlled by an adversary. This assumption allows for modelling non-stationary and highly stochastic reward signals. We will later show why our FLAD formulation fits into this setting. Under this setting, it is assumed that an adversary is given access to the learner's policy $\pi$ and determines the sequence of rewards, $(R_{a,t})_{t=1}^{N}$, for each arm prior to play [38]. At each turn $\pi$ determines a distribution over actions, $p(\mathcal{A})$, and an action is sampled from the distribution, $a \sim p(\mathcal{A})$. See Lattimore & Szepesvári [34] for further details.

**The EXP3 algorithm.** The EXP3 algorithm ("*Exp*onential-weight algorithm for *Exp*loration and *Exp*loitation") targets the adversarial multi-armed bandit problem by choosing arms according to a Gibbs distribution based on the empirically determined importance-weighted rewards of arms [21]. To allow for exploration, EXP3 mixes the Gibbs distribution with a uniform distribution.

Formally, let the exploration rate be $\gamma \in (0, 1]$. At round $t$, $\pi$ defines the probability of selecting a given arm, $a \in \mathcal{A}$, as a linear combination of Gibbs and uniform distributions

$$p_t(a) = (1 - \gamma) \frac{\exp(\gamma \hat{R}_{a,t-1}/K)}{\sum_{a'} \exp(\gamma \hat{R}_{a',t-1}/K)} + \frac{\gamma}{K} \tag{1}$$

where the importance weighted reward $\hat{R}_{a,t}$ is calculated as

$$\hat{R}_{a,t} = \hat{R}_{a,t-1} + \frac{R_{a,t}}{p_{t-1}(a)} \tag{2}$$

and $R_{a,t}$ denotes the observed reward. All unplayed arms, $a' \neq a$ have unchanged importance weighted rewards; $\hat{R}_{a',t} = \hat{R}_{a',t-1}$.

Algorithmically, EXP3 takes the following steps at each round: First, calculate the sampling distribution $p_t$ and sample an arm from the distribution. Then a reward $R_{a,t}$ is observed and the algorithm updates the importance weighted reward $\hat{R}_{a,t}$ for the played arm.

Informally, the use of an importance-weighted estimated reward compensates the rewards of actions that are less likely to be chosen, guaranteeing that the expected estimated reward is equal to the actual reward for each action. EXP3 is designed to be nearly optimal in the worst case, but due to the exploration rate it will select "bad" actions at a rate of $\gamma/K$. The exploration of EXP3 combined with importance-weighting allows the policy to handle non-stationary reward-generating processes.

**The UCB1 algorithm.** While the adversarial setting makes almost no assumptions about the reward-generating process and therefore maintains its performance guarantees under almost any circumstances, it can be outperformed in settings that *are* constrained. In this section we assume that the reward-generating processes are stationary Gaussian distributions. A common policy used to solve this MAB setting is the Upper Confidence Bound (UCB1) algorithm, which assigns each arm a value called the upper confidence bound based on Hoeffding's inequality [22]. The UCB1 algorithm is based on the principle of *optimism in the face of uncertainty*, meaning that with high probability the upper confidence bound assigned to each arm is an overestimate of the unknown mean reward.

Formally, let the estimated mean reward of arm $a$ after being played $n_a$ times be $\hat{R}_a$ and the true mean reward be $R_a$, then

$$\mathbb{P}\left(R_a \geq \hat{R}_a + \sqrt{\frac{2\ln(1/\delta)}{n_a}}\right) \leq \delta \quad \forall \delta \in (0, 1)$$

derived from Hoeffding's inequality (following equation 7.1 of Lattimore & Szepesvári [34]), where $\delta$ is the confidence level that quantifies the degree of certainty in the arm. In this work we let $\delta = 1/t$ where $t$ is the number of rounds played, shrinking the confidence bound over rounds. Thus, we define the upper confidence bound for arm $a$ at turn $t$ as

$$UCB_{a,t} = \begin{cases} \infty, & \text{if } n_a = 0 \\ \hat{R}_a + \sqrt{\frac{2\ln t}{n_a}}, & \text{otherwise} \end{cases} \tag{3}$$

Algorithmically, UCB1 takes the following steps at each round. First, the UCB1 policy plays the arm with largest upper confidence bound, $a^* = \arg\max_{a \in \mathcal{A}} UCB_{a,t}$. Next, a reward $R_{a^*,t}$ is observed and the algorithm updates $\hat{R}_{a^*}$ (the estimated mean reward for $a^*$) and the upper confidence bounds for all $a$. Informally, this algorithm suggests that the learner should play arms more often if they either 1. have large expected reward, $\hat{R}$, or 2. $n_a$ is small because the arm is not well explored.

# B  Pseudo-code

We include here pseudo-code for our 2 proposed algorithms. Algorithm 1 contains the pseudo-code for EXP3-FLAD, and Algorithm 2 contains the pseudo-code for UCB1-FLAD.

---

**Algorithm 1** EXP3-FLAD

---

**Require:** $\mathcal{D}_\mathcal{A}, \mathcal{D}_\mathcal{T}$: Auxiliary and target datasets
**Require:** $f_\theta$: Parameterized model
**Require:** $G$: Gradient accumulation steps
 1: **Initialize**: $K = |\mathcal{A}|$;   $\mathcal{E}_0 = \frac{1}{K}$;
        $\forall a \in \mathcal{A} : \nabla_a = 0, \hat{R}_a = 1$
 2: **for** $t = 1, 2, \ldots, N$ **do**
 3:    $\mathcal{E}_t = \min\left\{\frac{1}{K}, \sqrt{\frac{\ln K}{K \cdot t}}\right\}$
 4:    $\forall a \in \mathcal{A} : \pi(a) \leftarrow (1 - K\mathcal{E}_t)\frac{\exp(\mathcal{E}_{t-1}\hat{R}_a)}{\sum_{a'}\exp(\mathcal{E}_{t-1}R_{a'})} + \mathcal{E}_t$
 5:    Sample $a \sim \pi(\mathcal{A})$ and batch $\{\mathbf{x}, \mathbf{y}\} \sim \mathcal{D}_a$
 6:    $\nabla_a \leftarrow \nabla_a + \nabla_\theta \mathcal{L}(f_\theta, \mathbf{x}, \mathbf{y})$
 7:    **if** $t \pmod G \equiv 0$ **then**
 8:       $\nabla_\mathcal{T} \leftarrow \nabla_\theta \mathcal{L}(f_\theta, \mathcal{D}_\mathcal{T})$
 9:       Update model parameters w.r.t. $\nabla_\mathcal{T} + \sum_a \nabla_a$
10:       **for all** $\{a \in \mathcal{A} | \nabla_a \neq 0\}$ **do**
11:          $\hat{R}_a \leftarrow \hat{R}_a + \frac{R_{a,t}}{\pi(a)}$
12:          $\nabla_a \leftarrow 0$
13:       **end for**
14:    **end if**
15: **end for**

---

**Algorithm 2** UCB1-FLAD

---

**Require:** $\mathcal{D}_{\mathcal{A}}, \mathcal{D}_{\mathcal{T}}$: Auxiliary and target datasets
**Require:** $f_\theta$: Parameterized model
**Require:** $G$: Gradient accumulation steps
**Require:** $\beta$: Smoothing factor

1: **Initialize**:
$\quad \forall a \in \mathcal{A} : n_a = 1,$
$\qquad\qquad \hat{R}_a = \cos(\nabla_\theta \mathcal{L}(f_\theta, \mathcal{D}_{\mathcal{T}}), \nabla_\theta \mathcal{L}(f_\theta, \mathcal{D}_a))$
2: **for** $t = 1, 2, \ldots, N$ **do**
3: $\quad a^* = \underset{a \in \mathcal{A}}{\mathrm{argmax}} \ \hat{R}_a + \sqrt{\frac{2\ln t}{n_a}}$
4: $\quad$ Sample batch $\{\mathbf{x}, \mathbf{y}\} \sim \mathcal{D}_{a^*}$
5: $\quad \nabla_{a^*} \leftarrow \nabla_{a^*} + \nabla_\theta \mathcal{L}(f_\theta, \mathbf{x}, \mathbf{y})$
6: $\quad n_{a^*} \leftarrow n_{a^*} + 1$
7: $\quad$ **if** $t \ (\mathrm{mod} \ G) \equiv 0$ **then**
8: $\qquad \nabla_{\mathcal{T}} \leftarrow \nabla_\theta \mathcal{L}(f_\theta, \mathcal{D}_{\mathcal{T}})$
9: $\qquad$ Update model parameters w.r.t. $\nabla_{\mathcal{T}} + \sum_a \nabla_a$
10: $\qquad$ **for all** $\{a \in \mathcal{A} | \nabla_a \neq 0\}$ **do**
11: $\qquad\quad \hat{R}_a \leftarrow (1 - \beta)\hat{R}_a + \beta R_{a,t}$
12: $\qquad\quad \nabla_a \leftarrow 0$
13: $\qquad$ **end for**
14: $\quad$ **end if**
15: **end for**

---

## C    Training details

We train all models (FLAD and non-FLAD) on 40Gb A100s.

For all experiments, we use validation-based early stopping, and train for a maximum of 10,000 gradient update steps. In practice, we find that early-stopping leads to significantly fewer than 10,000 updates, usually between 50-150 for direct fine-tuning, and 1-2,000 for other methods.

For the smoothing factor, $\beta$, in UCB1-FLAD we ran preliminary experiments using values of $\{0.99, 0.9, 0.75, 0.5\}$ and found 0.9 to work well across datasets. All reported scores use $\beta = 0.9$.

In preliminary experiments we consider rewards using gradients from multiple model partitions: the full model, encoder-only, decoder-only, and language modelling head (token classifier). We find that using the parameters from the LM head provides best performance, followed by the decoder-only, encoder-only, and full model gradients. The differential from best to worst method was $\sim 3\%$ relative performance. Recall that with a gradient accumulation factor of $G$, our algorithms need to store at most $G + 1$ gradients at any time. So not only does using the LM head provide performance improvements, but also saves memory. For the models we use, the LM head contains only 2.3% of the full model parameters.

## D    Full results

The full results of experiments on target-only fine-tuning, explore-only, exploit-only, EXP3-FLAD, and UCB1-FLAD are found on the next page.

Table 2: Detailed results from the main experiment including direct fine-tuning, exploration-only, exploitation-only baselines and our proposed methods, EXP3-FLAD and UCB1-FLAD.

| Dataset | | Method | Anli-r1 | Anli-r2 | Anli-r3 | CB | COPA | HellaSwag | RTE | Story Cloze | WiC | Winogrande | WSC | Average |
|---|---|---|---|---|---|---|---|---|---|---|---|---|---|---|
| **T5-3B** | | Direct Fine-Tuning | 37.6 | 36.2 | 35.0 | 83.2 | 53.8 | 51.0 | 54.2 | 75.9 | 51.6 | 49.6 | 53.1 | 52.8 |
| | TOMix | Loss-Scaling ($GA$) | 35.7 | 36.4 | 35.3 | 82.5 | 58.0 | 51.8 | 59.0 | 79.6 | 49.8 | 50.4 | 46.9 | 53.2 |
| | | Loss-Scaling ($GMS$) | 37.8 | 37.6 | 36.0 | 80.0 | 76.4 | 52.6 | 55.3 | 85.7 | 50.6 | 52.0 | 51.7 | 56.0 |
| | | Exploration-Only | 38.1 | 40.3 | 36.7 | 88.6 | 85.6 | 51.2 | 67.6 | 88.8 | 51.0 | 55.5 | 47.7 | 59.2 |
| | | Exploitation-Only | 38.8 | 40.5 | 38.0 | 86.1 | 86.0 | 51.1 | 69.4 | 89.5 | 52.8 | 59.2 | 46.3 | 59.8 |
| | | EXP3-FLAD ($\mathcal{R}^{GA}$) | 40.6 | 39.9 | 36.9 | 86.1 | 89.8 | 52.0 | 76.7 | 90.8 | 50.5 | 60.3 | 52.9 | 61.5 |
| | | UCB1-FLAD ($\mathcal{R}^{GA}$) | 41.8 | 39.0 | 38.0 | 85.4 | 87.0 | 52.0 | 79.1 | 91.4 | 49.7 | 62.7 | 56.2 | 62.0 |
| | | EXP3-FLAD ($\mathcal{R}^{GMS}$) | 42.0 | 40.2 | 36.6 | 87.0 | 87.2 | 52.4 | 77.5 | 90.9 | 51.1 | 61.9 | 51.9 | 61.7 |
| | | UCB1-FLAD ($\mathcal{R}^{GMS}$) | 41.3 | 39.7 | 38.0 | 87.1 | 89.8 | 51.0 | 76.6 | 90.5 | 51.0 | 62.0 | 56.0 | 62.0 |
| | | EXP3-FLAD ($\mathcal{R}^{AGG}$) | 38.6 | 39.8 | 39.1 | 86.8 | 91.2 | 51.2 | 78.8 | 90.4 | 50.7 | 63.0 | 52.9 | 62.0 |
| | | UCB1-FLAD ($\mathcal{R}^{AGG}$) | 42.0 | 41.0 | 36.6 | 88.2 | 86.8 | 51.0 | 77.3 | 90.3 | 51.1 | 63.3 | 55.4 | 62.1 |
| | P3 | Loss-Scaling ($GA$) | 38.7 | 39.5 | 34.8 | 80.7 | 64.4 | 52.7 | 62.9 | 80.1 | 50.3 | 51.9 | 51.2 | 55.2 |
| | | Loss-Scaling ($GMS$) | 39.2 | 38.7 | 36.4 | 85.0 | 67.8 | 51.9 | 62.4 | 84.8 | 50.3 | 51.8 | 52.1 | 56.4 |
| | | Exploration-Only | 40.1 | 37.7 | 36.0 | 85.4 | 83.6 | 52.1 | 77.3 | 89.1 | 51.5 | 57.2 | 57.1 | 60.6 |
| | | Exploitation-Only | 40.4 | 37.2 | 37.3 | 87.1 | 84.4 | 51.0 | 78.6 | 90.3 | 51.3 | 56.2 | 51.5 | 60.5 |
| | | EXP3-FLAD ($\mathcal{R}^{GA}$) | 46.9 | 38.8 | 40.2 | 89.6 | 88.0 | 51.5 | 76.9 | 91.2 | 53.4 | 66.2 | 61.9 | 64.1 |
| | | UCB1-FLAD ($\mathcal{R}^{GA}$) | 49.1 | 40.1 | 40.1 | 88.6 | 88.2 | 51.6 | 83.7 | 90.2 | 54.3 | 68.0 | 68.3 | 65.5 |
| | | EXP3-FLAD ($\mathcal{R}^{GMS}$) | 46.2 | 40.6 | 39.4 | 88.9 | 90.4 | 51.6 | 85.1 | 91.3 | 54.4 | 65.8 | 67.5 | 65.6 |
| | | UCB1-FLAD ($\mathcal{R}^{GMS}$) | 48.1 | 40.1 | 39.1 | 87.5 | 89.4 | 52.0 | 83.7 | 89.4 | 51.7 | 70.6 | 70.6 | 65.6 |
| | | EXP3-FLAD ($\mathcal{R}^{AGG}$) | 47.6 | 40.6 | 40.6 | 90.0 | 90.6 | 51.4 | 84.5 | 91.0 | 53.2 | 66.7 | 64.0 | 65.5 |
| | | UCB1-FLAD ($\mathcal{R}^{AGG}$) | 42.0 | 41.0 | 41.2 | 86.8 | 86.8 | 51.5 | 85.5 | 91.1 | 52.7 | 66.3 | 70.6 | 65.6 |
| **T0-3B** | | Direct Fine-Tuning | 40.9 | 39.1 | 37.1 | 79.6 | 66.4 | 43.5 | 67.1 | 83.2 | 52.5 | 54.6 | 56.7 | 56.4 |
| | TOMix | Loss-Scaling ($GA$) | 41.3 | 40.0 | 36.9 | 81.8 | 78.0 | 51.2 | 76.5 | 86.9 | 50.7 | 54.7 | 56.2 | 59.5 |
| | | Loss-Scaling ($GMS$) | 40.5 | 40.5 | 37.8 | 81.1 | 79.0 | 52.0 | 77.0 | 88.8 | 52.7 | 55.0 | 60.8 | 60.5 |
| | | Exploration-Only | 44.4 | 40.3 | 37.0 | 82.5 | 85.6 | 47.9 | 77.6 | 90.1 | 52.1 | 58.6 | 56.9 | 61.2 |
| | | Exploitation-Only | 42.5 | 39.3 | 37.2 | 84.3 | 82.8 | 48.1 | 79.7 | 88.8 | 52.8 | 57.8 | 56.3 | 60.9 |
| | | EXP3-FLAD ($\mathcal{R}^{GA}$) | 46.2 | 41.5 | 37.7 | 83.9 | 87.6 | 49.4 | 80.0 | 90.1 | 52.6 | 63.4 | 59.0 | 62.9 |
| | | UCB1-FLAD ($\mathcal{R}^{GA}$) | 43.7 | 40.8 | 37.6 | 86.1 | 85.4 | 48.6 | 80.5 | 91.3 | 53.4 | 63.5 | 61.0 | 62.9 |
| | | EXP3-FLAD ($\mathcal{R}^{GMS}$) | 43.4 | 41.1 | 38.2 | 84.6 | 86.6 | 49.1 | 81.0 | 90.6 | 53.0 | 63.1 | 59.8 | 62.8 |
| | | UCB1-FLAD ($\mathcal{R}^{GMS}$) | 43.2 | 41.2 | 38.7 | 86.4 | 86.6 | 48.4 | 82.8 | 91.4 | 52.2 | 65.8 | 59.4 | 62.8 |
| | | EXP3-FLAD ($\mathcal{R}^{AGG}$) | 43.8 | 41.6 | 38.0 | 86.4 | 87.8 | 48.9 | 81.9 | 90.7 | 52.5 | 66.7 | 59.8 | 62.8 |
| | | UCB1-FLAD ($\mathcal{R}^{AGG}$) | 44.0 | 41.6 | 38.3 | 85.4 | 87.4 | 48.6 | 81.1 | 90.6 | 53.0 | 66.3 | 59.2 | 62.9 |
| | P3 | Loss-Scaling ($GA$) | 44.0 | 40.4 | 38.9 | 86.4 | 77.6 | 51.0 | 75.1 | 86.8 | 51.7 | 55.6 | 59.8 | 60.7 |
| | | Loss-Scaling ($GMS$) | 43.8 | 38.6 | 39.3 | 82.5 | 79.2 | 50.6 | 80.6 | 89.1 | 51.6 | 56.6 | 56.0 | 60.7 |
| | | Exploration-Only | 45.4 | 40.3 | 38.0 | 82.5 | 87.8 | 50.6 | 82.2 | 88.8 | 52.4 | 61.8 | 60.6 | 62.8 |
| | | Exploitation-Only | 45.5 | 40.0 | 38.8 | 87.5 | 82.2 | 49.9 | 79.6 | 90.9 | 52.2 | 60.1 | 64.8 | 62.9 |
| | | EXP3-FLAD ($\mathcal{R}^{GA}$) | 50.4 | 40.0 | 41.2 | 87.9 | 88.4 | 49.7 | 86.1 | 91.6 | 52.8 | 67.5 | 70.4 | 66.0 |
| | | UCB1-FLAD ($\mathcal{R}^{GA}$) | 48.2 | 41.8 | 41.8 | 90.0 | 86.6 | 50.0 | 86.1 | 91.5 | 53.6 | 65.6 | 74.6 | 66.3 |
| | | EXP3-FLAD ($\mathcal{R}^{GMS}$) | 49.5 | 40.8 | 39.5 | 87.1 | 89.2 | 49.4 | 85.8 | 91.4 | 53.9 | 65.4 | 68.7 | 65.5 |
| | | UCB1-FLAD ($\mathcal{R}^{GMS}$) | 48.2 | 41.8 | 41.2 | 85.4 | 88.0 | 49.6 | 83.2 | 91.6 | 52.6 | 66.1 | 74.6 | 66.0 |
| | | EXP3-FLAD ($\mathcal{R}^{AGG}$) | 51.1 | 40.3 | 39.9 | 89.6 | 91.4 | 49.0 | 86.5 | 91.6 | 52.6 | 66.4 | 76.7 | 66.8 |
| | | UCB1-FLAD ($\mathcal{R}^{AGG}$) | 49.8 | 39.9 | 40.8 | 86.8 | 88.4 | 49.6 | 84.7 | 91.0 | 53.2 | 68.0 | 76.9 | 66.3 |

# E    Probing the reward generating processes.

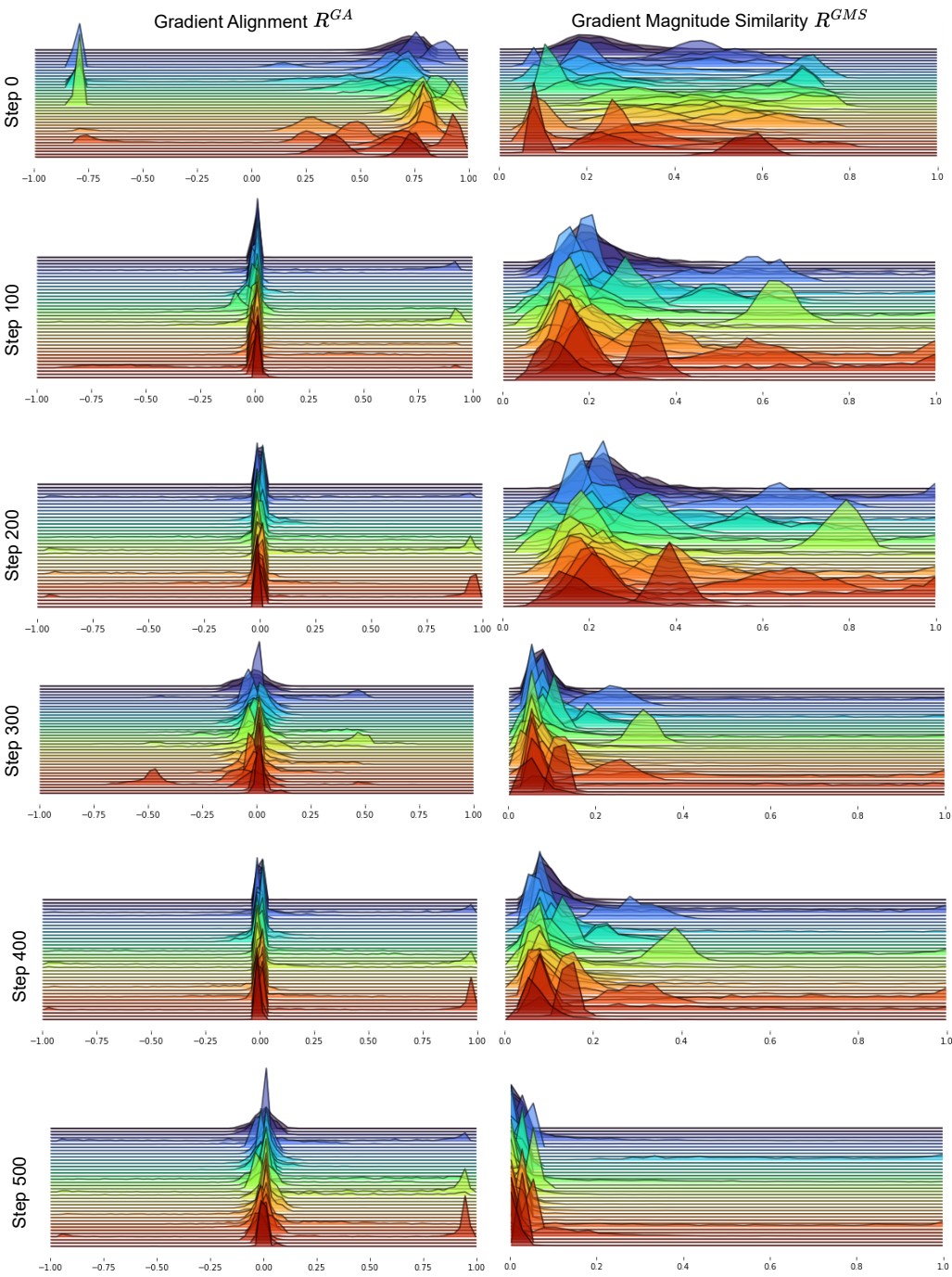

Figure 5: **Reward distributions** of $\mathcal{R}^{GA}$ and $\mathcal{R}^{GMS}$ prior to training and every 100 gradient updates thereafter. We probe the reward distributions using the T5-XL model with the T0Mix auxiliary dataset and WSC [52] as the target dataset.

# F    EXP3-FLAD and UCB1-FLAD training dynamics

The following 4 pages include a case study on the training dynamics of EXP3-FLAD and UCB1-FLAD when training T5-XL using T0Mix as the auxiliary data. First, we find datasets where EXP3-FLAD and UCB1-FLAD improve significantly over the baseline FLAD methods, but also where either EXP3-FLAD or UCB1-FLAD clearly outperforms the other. The two datasets that fulfill our interests are RTE and COPA.

We find that UCB1-FLAD outperforms EXP3-FLAD on RTE, and show their respective training dynamics in Figure 6 (UCB1) and Figure 7 (EXP3).

We find that EXP3-FLAD outperforms UCB1-FLAD on COPA, and show their respective training dynamics in Figure 8 (UCB1) and Figure 9 (EXP3).

We include details and takeaways in the caption for each figure. For EXP3-FLAD figures, we include charts of the cumulative estimated reward, empirical gradient alignment, instantaneous sampling distribution determined by the policy, and the empirical sampling distribution determined by the total number of samples seen per dataset as a fraction of the total samples seen. For UCB1-FLAD figures, we include charts of the upper confidence index, estimated gradient alignment, and the empirical sampling distribution.

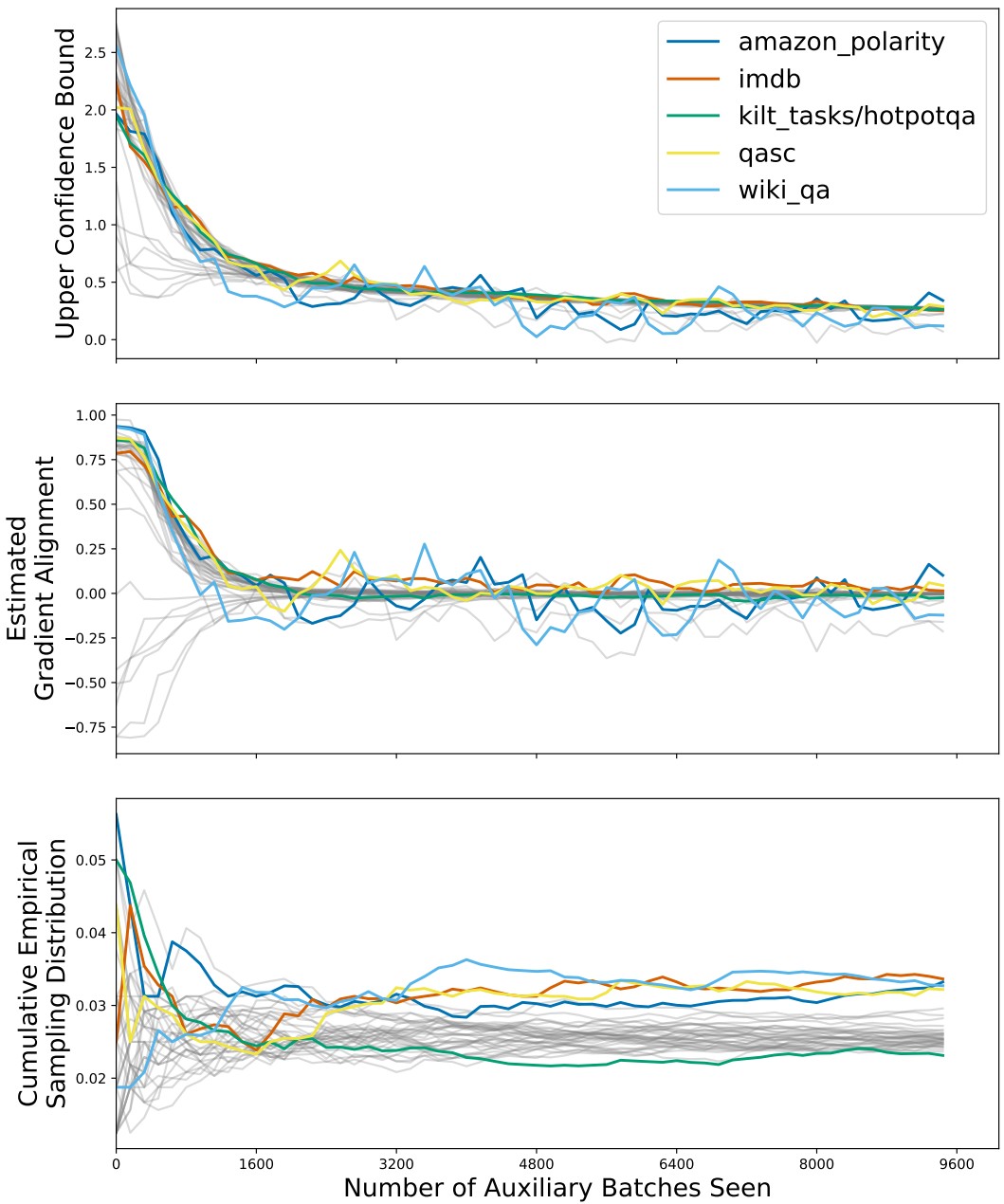

Figure 6: Training dynamics of UCB1-FLAD, a case study using RTE as target dataset and T0Mix as auxiliary data, where UCB1-FLAD outperforms EXP3-FLAD. Colored lines are a sample of auxiliary datasets with interesting properties, the remaining datasets are shown in grey. We find that even though wiki_qa's estimated gradient alignment falls to below 0 (middle), UCB1 does not abandon sampling from it in the future, finding that between 3200 and 4800 batches, it becomes the dataset with largest upper confidence bound (top). Similarly, we see that UCB1 alternates between wiki_qa, amazon_polarity, and qasc as the datasets with higher gradient alignment and upper confidence bounds. kilt_tasks/hotpotqa has a very high gradient alignment prior to training, but UCB1 samples very infrequently from it, due to it'ls lower upper confidence bound. This is a failure case for transfer learning-based methods. Interestingly, UCB1 never estimates imdb to have a negative gradient, and gradually samples from it more and more frequently over the course of training.

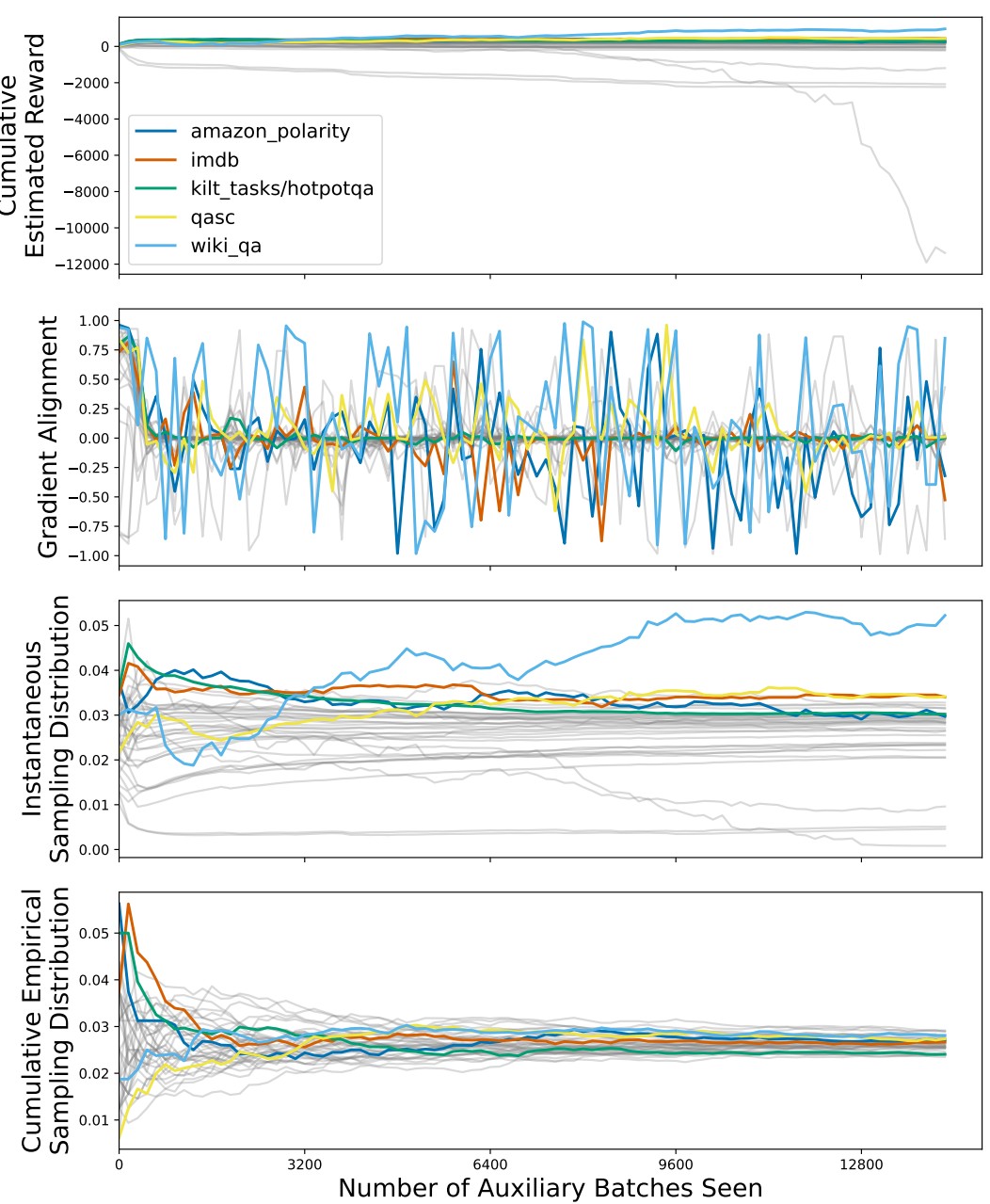

Figure 7: Training dynamics of EXP3-FLAD, a case study using RTE as target dataset and T0Mix as auxiliary data, where UCB1-FLAD outperforms EXP3-FLAD. Colored lines are a sample of auxiliary datasets with interesting properties, the remaining datasets are shown in grey. We find that the gradient alignment signal is particularly noisy for EXP3-FLAD, possibly leading to it's slightly worse performance on RTE. All five highlighted auxiliary datasets have high instantaneous sampling probability, but over the course of training, the empirical sampling distribution is very condensed across the full set of auxiliary datasets, unlike UCB1 which is able to find better separation.

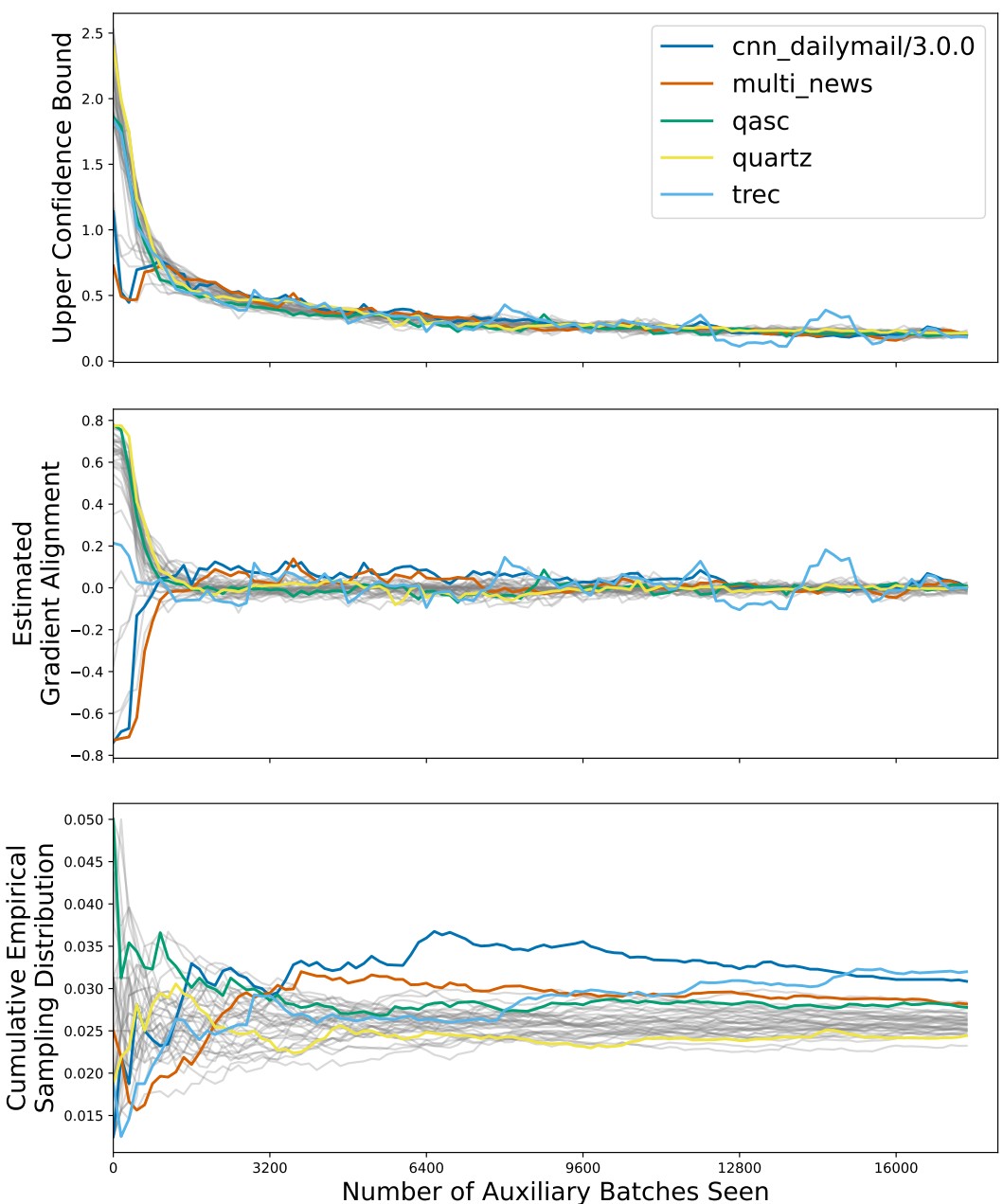

Figure 8: Training dynamics of UCB1-FLAD, a case study using COPA as target dataset and T0Mix as auxiliary data, where EXP3-FLAD outperforms UCB1-FLAD. Colored lines are a sample of auxiliary datasets with interesting properties, the remaining datasets are shown in grey. We find that although qasc and quartz start with very high gradient alignment, they very quickly fall to negative alignment (middle figure, green and yellow). In the end, we find that the algorithm samples much more from qasc than from quartz (bottom figure). Interestingly, we find that although both cnn_dailymail and multi_news start off with very negative gradient alignment, they quickly become the most aligned with the target task (middle figure, blue and red). We find that the three auxiliary datasets with highest upper confidence index (top figure) and largest sampling percent (bottom figure) are cnn_dailymail, multi_news, and trec even though these all considered dissimilar to the target prior to training.

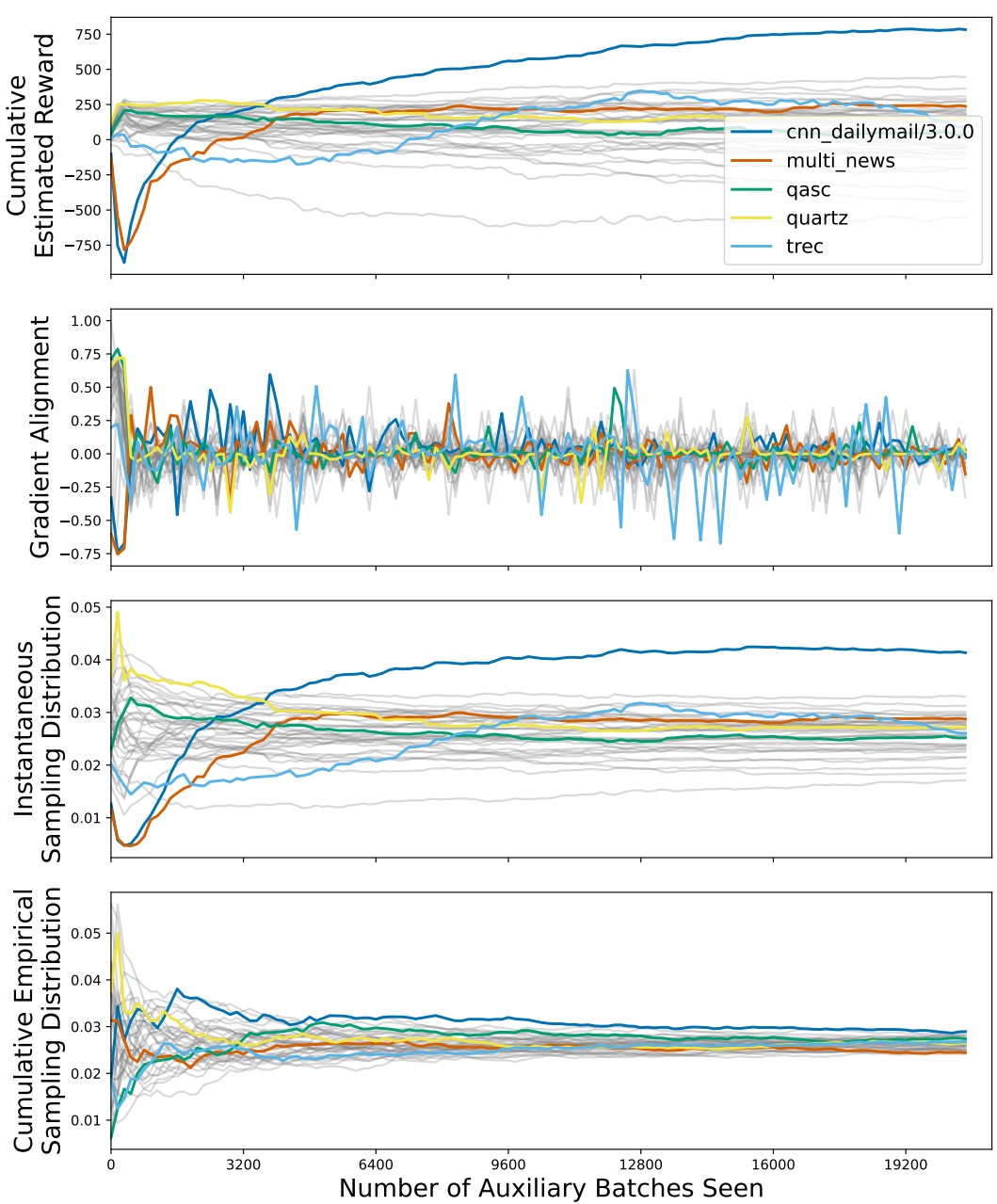

Figure 9: Training dynamics of EXP3-FLAD, a case study using COPA as target dataset and T0Mix as auxiliary data, where EXP3-FLAD outperforms UCB1-FLAD. Colored lines are a sample of auxiliary datasets with interesting properties, the remaining datasets are shown in grey. This is an impressive example of the importance-weighted estimated reward. We see that cnn_dailymail and multi_news both start with very negative alignment, but EXP3 quickly updates it's estimated reward once their alignment becomes positive. Similar to RTE, we see that EXP3 never makes large separations in the empirical sampling distribution, possibly a reason why UCB1 outperforms EXP3 overall. Compared to RTE, we find that gradient alignments are much less variable, with a maximum alignment close to 0.5 and minimum alignment close to -0.5. Whereas in RTE, alignments regularly reach close to 1.0 and -1.0.

|  | Explore-Only | Exploit-Only | EXP3-FLAD ($\mathcal{R}^{GA}$) | UCB1-FLAD ($\mathcal{R}^{GA}$) |
|---|---|---|---|---|
| $|\mathcal{A}| = 35$ (T0Mix) | 570.9 | 549.1 | 769.1 | 700.0 |
| $|\mathcal{A}| = 260$ (P3) | 863.6 | 692.7 | 832.7 | 794.5 |
| % increase | 51.3% | 26.2% | 8.3% | 13.5% |

Table 3: Number of training iterations for T0-3B to converge using a training method (column) and a set of auxiliary datasets (row). The number of iterations to convergence is averaged across 11 target datasets and 5 seeds, leading to 55 experiments aggregated per cell.

## G  Effect of scaling $|\mathcal{A}|$ on time-to-convergence

As we have described in this work, the computational complexity for a single turn of our methods are independent of the number of auxiliary datasets. However, it is unclear whether the computation complexity of the multi-armed bandits are dependent on the number of auxiliary datasets through their exploration rates. Thus, the computational complexity of an individual training run may be influenced by the number of auxiliary datasets ($|\mathcal{A}|$), but it is not possible to characterize this relation explicitly as it relates to the complex and stochastic process of training and large language model.

To better understand the empirical effects of increasing $|\mathcal{A}|$ on the time-to-model-convergence, we perform a study on the number of iterations to convergence for different FLAD algorithms. Table 3 shows that all methods require longer training to converge when increasing from $|\mathcal{A}| = 35$ to $260$. We find that, compared with baseline methods, our MAB-based methods require more steps for the smaller set of auxiliary datasets, but the number of additional steps required to train our methods only increases modestly ($\sim 10\%$) when increasing $|\mathcal{A}|$ by a factor of nearly 10. In contrast, the Explore- and Exploit-Only methods do not scale nearly as well when increasing the number of auxiliary datasets. Notably, the Explore-Only method requires over 50% more training iterations for P3 than for T0Mix, at which point it takes longer to converge than either of the MAB-based methods.

## H  Auxiliary Datasets

Here we include the full list of auxiliary datasets from P3 [23] used to train models for the ANLI target tasks. Other target datasets have slightly different auxiliary datasets due to test set decontamination, but are generally the same. Datasets are listed by their name as found in HuggingFace Datasets[2].

Zaid/quac_expanded, acronym_identification, ade_corpus_v2/Ade_corpus_v2_classification, ade_corpus_v2/Ade_corpus_v2_drug_ade_relation, ade_corpus_v2/Ade_corpus_v2_drug_dosage_relation, adversarial_qa/adversarialQA, adversarial_qa/dbert, adversarial_qa/dbidaf, adversarial_qa/droberta, aeslc, ag_news, ai2_arc/ARC-Challenge, ai2_arc/ARC-Easy, amazon_polarity, amazon_reviews_multi/en, amazon_us_reviews/Wireless_v1_00, ambig_qa/light, app_reviews, aqua_rat/raw, art, asset/ratings, asset/simplification, banking77, billsum, bing_coronavirus_query_set, biosses, blbooksgenre/title_genre_classifiction, blended_skill_talk, cbt/CN, cbt/NE, cbt/P, cbt/V, cbt/raw, cc_news, circa, climate_fever, cnn_dailymail/3.0.0, codah/codah, codah/fold_0, codah/fold_1, codah/fold_2, codah/fold_3, codah/fold_4, code_x_glue_tc_text_to_code, common_gen, commonsense_qa, conv_ai, conv_ai_2, conv_ai_3, cord19/metadata, cos_e/v1.0, cos_e/v1.11, cosmos_qa, covid_qa_castorini, craffel/openai_lambada, craigslist_bargains, crows_pairs, dbpedia_14, discofuse/discofuse-sport, discofuse/discofuse-wikipedia, discovery/discovery, docred, dream, drop, duorc/ParaphraseRC, duorc/SelfRC, e2e_nlg_cleaned, ecthr_cases/alleged-violation-prediction, emo, emotion, enriched_web_nlg/en, esnli, evidence_infer_treatment/1.1, evidence_infer_treatment/2.0, fever/v1.0, fever/v2.0, financial_phrasebank/sentences_allagree, freebase_qa, generated_reviews_enth, gigaword, glue/ax, glue/cola, glue/mnli, glue/mnli_matched, glue/mnli_mismatched, glue/mrpc, glue/qnli, glue/qqp, glue/rte, glue/sst2, glue/stsb, glue/wnli, google_wellformed_query, great_code, guardian_authorship/cross_genre_1, guardian_authorship/cross_topic_1, guardian_authorship/cross_topic_4, guardian_authorship/cross_topic_7, gutenberg_time, hans, hate_speech18, head_qa/en, health_fact, hlgd, hotpot_qa/distractor, hotpot_qa/fullwiki,

---

[2]https://huggingface.co/datasets

humicroedit/subtask-1, humicroedit/subtask-2, hyperpartisan_news_detection/byarticle, hyperpartisan_news_detection/bypublisher, imdb, jfleg, kelm, kilt_tasks/hotpotqa, kilt_tasks/nq, lama/trex, lambada, liar, limit, math_dataset/algebra__linear_1d, math_dataset/algebra__linear_1d_composed, math_dataset/algebra__linear_2d, math_dataset/algebra__linear_2d_composed, math_qa, mc_taco, mdd/task1_qa, mdd/task2_recs, mdd/task3_qarecs, medal, medical_questions_pairs, meta_woz/dialogues, mocha, movie_rationales, multi_news, multi_nli, multi_x_science_sum, mwsc, narrativeqa, ncbi_disease, neural_code_search/evaluation_dataset, newspop, nlu_evaluation_data, nq_open, numer_sense, onestop_english, openai_humaneval, openbookqa/additional, openbookqa/main, paws-x/en, paws/labeled_final, paws/labeled_swap, paws/unlabeled_final, piqa, poem_sentiment, pubmed_qa/pqa_labeled, qa_srl, qa_zre, qasc, qed, quac, quail, quarel, quartz, quora, quoref, race/all, race/high, race/middle, riddle_sense, ropes, rotten_tomatoes, samsum, scan/addprim_jump, scan/addprim_turn_left, scan/filler_num0, scan/filler_num1, scan/filler_num2, scan/filler_num3, scan/length, scan/simple, scan/template_around_right, scan/template_jump_around_right, scan/template_opposite_right, scan/template_right, scicite, scientific_papers/arxiv, scientific_papers/pubmed, sciq, scitail/snli_format, scitail/tsv_format, scitldr/Abstract, selqa/answer_selection_analysis, sem_eval_2010_task_8, sem_eval_2014_task_1, sent_comp, sick, sms_spam, snips_built_in_intents, snli, social_i_qa, species_800, squad, squad_adversarial/AddSent, squad_v2, squadshifts/amazon, squadshifts/new_wiki, squadshifts/nyt, sst/default, stsb_multi_mt/en, subjqa/books, subjqa/electronics, subjqa/grocery, subjqa/movies, subjqa/restaurants, subjqa/tripadvisor, super_glue/axb, super_glue/axg, super_glue/boolq, super_glue/multirc, super_glue/record, swag/regular, tab_fact/tab_fact, tmu_gfm_dataset, trec, trivia_qa/unfiltered, turk, tweet_eval/emoji, tweet_eval/emotion, tweet_eval/hate, tweet_eval/irony, tweet_eval/offensive, tweet_eval/sentiment, tweet_eval/stance_abortion, tweet_eval/stance_atheism, tweet_eval/stance_climate, tweet_eval/stance_feminist, tweet_eval/stance_hillary, tydiqa/primary_task, tydiqa/secondary_task, web_questions, wiki_bio, wiki_hop/masked, wiki_hop/original, wiki_qa, wiki_split, wino_bias/type1_anti, wino_bias/type1_pro, wino_bias/type2_anti, wino_bias/type2_pro, winograd_wsc/wsc273, winograd_wsc/wsc285, wiqa, xnli/en, xquad/xquad.en, xquad_r/en, xsum, yahoo_answers_qa, yahoo_answers_topics, yelp_polarity, yelp_review_full, zest

