# OpenReview forum: "Improving Few-Shot Generalization by Exploring and Exploiting Auxiliary Data"
_NeurIPS.cc/2023/Conference — NeurIPS 2023 poster_

### Official Review · Reviewer_5GTn · 2023-07-03

**Soundness:** 3 good
**Presentation:** 3 good
**Contribution:** 3 good
**Rating:** 7
**Confidence:** 2

**Summary:**

The paper presents a new approach to Few-shot Learning with Auxiliary Data (FLAD), a method that uses additional data to improve model generalization. Traditional methods for integrating auxiliary and target data have limitations, particularly in terms of computational scale with an increasing number of auxiliary datasets. This study introduces two new algorithms, EXP3-FLAD and UCB1-FLAD, that relate FLAD to the explore-exploit problem in multi-armed bandit scenarios, thereby achieving computational complexity that is independent of the quantity of auxiliary datasets. This paper shows that a combined exploration and exploitation strategy is vital, with the new methods surpassing all existing FLAD methods by 4%. The research also yielded the first 3 billion parameter language models that outperform the 175 billion parameters GPT-3.

**Strengths:**

* This study introduces a novel approach to enhance few-shot learning in language models by leveraging auxiliary data. The innovative use of additional datasets significantly improves the generalization capabilities of the models in the few-shot learning scenario.

* The incorporation of an exploration/exploitation framework, often seen in multi-armed bandit scenarios, adds an interesting aspect to the methodology. It ensures the computational complexity is independent of the number of auxiliary datasets, thus addressing the significant issue of scalability often seen in previous methodologies.

* The experimental results displayed in Table 1 and Figure 2 are good and offer strong support for the efficacy of this approach.

* One of the most exciting features of this methodology is its universal applicability; it can be effectively utilized with any language model. It provides a path to significantly elevate the performance of relatively smaller models, leveraging them to outperform even the more sizable models, as demonstrated by the 3 billion parameter language model surpassing the 175 billion parameters GPT-3.

**Weaknesses:**

* Figure 2 presents an interesting picture of the model's performance, revealing an interesting dependency on the auxiliary data for some tasks, while seeming almost indifferent between using T0 or T5 for others. The interpretation of these results is not straightforward and questions about the behavior exhibited by the model across different tasks.

* In terms of its responsiveness to auxiliary data and the use of T0 or T5, the performance differs significantly from task to task. This variance potentially suggests that certain tasks might be more intrinsically aligned with the structure or features of the auxiliary data, or that some tasks may respond more favorably to the model architectures embodied by T0 and T5, but there is no ablation or discussion about this aspect.

**Questions:**

* Why is the policy configured to select a singular specific auxiliary dataset? Could there not be more value in a more flexible approach, allowing the policy to identify and utilize the most appropriate samples across the entirety of the auxiliary data? Especially considering the successes achieved with cross-task transfer for T5 and T0, it seems that an exploration of this approach could be beneficial.

* Figure 2 the HellaSwag task, where all methods show inferior performance, with the exception of GPT-3. Could there be certain features inherent to this task that align better with the GPT-3 model, or are there particular aspects of the task that the other models struggle with? Do you have some intuition?

* Do you have updated results against GPT-4?

**Limitations:**

* I do not have much practical experience with RL and it is difficult for me to judge the limitations of this method.

---

> ### Author Rebuttal · Authors · 2023-08-10
>
> We thank the reviewer for their insightful questions and overall appreciation of our work, we share an excitement about the proposed methodology.
>
> >Figure 2 presents an interesting picture of the model's performance, revealing an interesting dependency on the auxiliary data for some tasks, while seeming almost indifferent between using T0 or T5 for others. ...
>
> We’d like to clarify that all four of our models in Figure 2 were trained with the same auxiliary data, P3. The difference between each model is the algorithm used in FLAD training (EXP3-FLAD or UCB1-FLAD), and the base model used to initialize FLAD training (T5 or T0).
>
> Figure 2 presents a very clear separation in performance for two datasets: anli-r1 and wsc (T0-based models have significantly better performance than T5). The fact that we *only* see such a separation in 2/11 datasets was a surprising result for us, as we expected that the multitask-trained T0 would always be a better starting point for our FLAD training. The exact reason why starting with T0 only improves performance on these two datasets is difficult to determine. Because the models in Figure 2 were all trained on P3, but only T0 has been multitask trained explicitly on T0Mix, these results suggest anli-r1 and wsc benefit from the extensive multitask training of T0. Conversely, this suggests that for the remaining 9/11 target datasets, our FLAD methods influence few-shot generalization ability more than general-purpose multitask training.
>
> Thank you for pointing out this interesting direction of analysis, we will add this surprising result to the analysis of the paper.
>
> >In terms of its responsiveness to auxiliary data and the use of T0 or T5, the performance differs significantly from task to task. ...
>
> We agree that the variance across tasks is likely coming from the available auxiliary data, but should note that there is no architectural difference between T5 and T0, so we don’t believe that architectural differences could cause the variations in performance.
>
> Figure 3 demonstrates how certain auxiliary tasks are more aligned with the features of the WSC target task, and In particular we show that at step 300 there is a clear differentiation between those auxiliary datasets which we believe to be aligned with the target task (Figure 4 in the appendix expands this figure to every 100 training steps).
>
> Additionally, Section F in the Appendix shows four figures containing in-depth information on the alignment between target and auxiliary tasks. Specifically, consider Figure 5 in the appendix. We look at the training dynamics of the target task RTE, which is a binary classification task. The figure shows that amazon_polarity, imdb, wiki_qa, qasc are the auxiliary datasets which are most aligned with RTE, and thus most sampled datasets. Interestingly, these tasks are all binary classification, or multiple choice datasets formulated as binary classification. On the other hand, we see that kilt_tasks/hotpotqa, which is a generation task, is the least sampled dataset. These examples suggest that the structure of the auxiliary datasets are, at least partially, a predictor of usefulness for target tasks.
>
> We will add these interesting findings into the discussion section of the main paper.
>
> >Why is the policy configured to select a singular specific auxiliary dataset? Could there not be more value in a more flexible approach, allowing the policy to identify and utilize the most appropriate samples across the entirety of the auxiliary data? ...
>
> First, there is definitely value in applying approaches that identify and utilize the most appropriate samples across the full set of auxiliary data! In fact, there is an excellent direction of study called data selection that focuses on exactly that question (we briefly discuss this area and provide references in the related works section on lines 89-92).
>
> Second, we actually compare against DEFT-Few (Figure 2) which does something similar to what you are suggesting. In DEFT-Few they train their model on the 500 nearest neighbor auxiliary samples to the target dataset (more detail on lines 290-293). We found that our method outperforms DEFT-Few by 2.7% absolute accuracy under the same settings (T0 as the base model, P3 as the auxiliary data).
>
> Finally, our method calculates the policy online (at every turn) and there are some downsides to identifying the most appropriate samples across the entirety of the auxiliary data at every turn. Mainly, the method would need to reindex the entirety of the auxiliary data at every turn (or every N turns), making it very costly and inefficient. In contrast, to update our policy, we only need to know the reward from the current turn, making it very efficient.
>
> >Figure 2 the HellaSwag task, ...
>
> Thank you for pointing this out, it’s a known phenomena that is difficult to give an exact answer to because the exact details of GPT-3 training is unknown. Sanh et al. [1] and Wei et al. [2] also found that their models (T0 and FLAN) underperform GPT-3 on HellaSwag. They suggest that because this task can be formulated as finishing an incomplete sentence, it is ideally formulated for large language models such as GPT-3. Furthermore, we note that HellaSwag was created through adversarial filtering [3], leading to a dataset with adversarially challenging samples and may adversely affect smaller models more than large models.
>
> >Do you have updated results against GPT-4?
>
> We assume the performance of GPT-4 would be very strong in this setting, however, in this work we focus on methods of achieving high performance from much smaller base models.
>
> [1] - Sanh et al. Multitask Prompted Training Enables Zero-Shot Task Generalization, 2023. https://arxiv.org/abs/2110.08207
>
> [2] - Wei et al. Finetuned Language Models are Zero-Shot Learners, 2022. https://arxiv.org/abs/2109.01652
>
> [3] Zellers et al. HellaSwag: Can a Machine Really Finish Your Sentence?, 2019. https://arxiv.org/abs/1905.07830

---

> > ### Comment · Reviewer_5GTn · 2023-08-13
> >
> > Thank you for the rebuttal. It clarified my questions.

---

### Official Review · Reviewer_jiND · 2023-07-03

**Soundness:** 3 good
**Presentation:** 2 fair
**Contribution:** 3 good
**Rating:** 6
**Confidence:** 3

**Summary:**

The paper addresses the problem of using auxiliary  data for the few-shot learning. The main novelty of the proposed approach is to formulate the selection of the auxiliary dataset from a pool of datasets as a multi-arm bandit problem (balancing exploration and exploitation) and adapting EXP3 and UCB1 to that goal. This allows to scale the number axillary datasets and improve generalization in the few-shot learning. The method is evaluated in application to NLP.

**Strengths:**

Important problem
Improves the result with a smaller model
Shows nice empirical results

**Weaknesses:**

The novelty of the paper is incremental
There are some problems in the presentation that cause confusion as explained below.

**Questions:**

One of the main claims in the paper is that the proposed approach almost doesn't add to computational cost.  The  description in line 172-174: " This is done for each auxiliary dataset by calculating the gradient ∇a = ∇θL(fθ, x, y), where the number of samples in {x, y} is significantly larger than a mini-batch,  and can be up to the size of the full dataset."  doesn't seem to go along with that clain, as this computation is large and it has to be done in every round. Do I miss something here?

A more minor thing that needs to be taken care of is in line 219. Exploit-only is suppose to be an opposite to Explore-only. So it's confusing how Exploit-only is an extension of Explore-Only. The text later explains the idea, but it needs amendment.

**Limitations:**

I did not really see any clear discussion of limitations. While the selection of the datasets are done using the proposed algorithms of balancing exploration and exploitation, there is still an implicit assumption that the datasets are useful. I believe that in language tasks this is less of a problem but in general it could be the case.

---

> ### Author Rebuttal · Authors · 2023-08-10
>
> Thank you for your very insightful questions and for the opportunity to respond with the clarifications below:
>
> >The novelty of the paper is incremental
>
> We would like to clarify what we believe is an important misunderstanding. While the building blocks of our algorithms may be derived from existing algorithms, we emphasize that the ultimate impact of research often lies in how the components are combined and applied, and the resulting effectiveness. Our work presents a novel application of the multi-armed bandit framework, and demonstrates a significant advancement of methods in the Few-shot Learning with Auxiliary Data (FLAD) setting.
>
> Our novel algorithms (EXP3-FLAD and UCB1-FLAD) are shown to solve an important problem within FLAD, namely how to scale to the ever-increasing number of available datasets. Through our adaptation of MAB to the FLAD setting, we show that we can scale to 100x more auxiliary datasets than previous methods, leading to a 4% accuracy improvement. These results mark a leap in the efficiency and effectiveness of few-shot learning, which can hardly be considered incremental. Thus, while we have built upon existing techniques from multi-armed bandits, we believe the outcome is a transformative contribution to the field of few-shot learning.
>
> Additionally, we would like to emphasize that it was not obvious, a priori, that our proposed methods would work. In particular, recall from lines 202-204 that we divide each few-shot target dataset into equal training and validation partitions, where the training partition is used to calculate rewards and update model parameters, while the validation partition is used for early stopping and selecting the best model checkpoint. One particularly novel and unexpected finding is that even with only 10-35 training samples, our reward functions are able to create a meaningful signal for the MAB algorithms.
>
> >One of the main claims in the paper is that the proposed approach almost doesn't add to computational cost. The description in line 172-174: " This is done for each auxiliary dataset by calculating the gradient $\nabla a=\nabla L(f_{\theta},x,y)$, where the number of samples in {x, y} is significantly larger than a mini-batch, and can be up to the size of the full dataset." doesn't seem to go along with that clain, as this computation is large and it has to be done in every round. Do I miss something here?
>
> We would like to clarify that the computation you are referring to is **not** performed every round. As described in lines 170-171, this computation is used to initialize the estimated reward for each arm prior to training, thus it only occurs once. To provide you with a more detailed answer, in traditional UCB1 the algorithm starts by playing each arm for one turn (to initialize rewards), and then continues accordingly. In our work, we take advantage of the fact that we can obtain a better estimate of the initial reward by using a larger number of samples prior to training (rather than a single batch as would be done if we strictly followed UCB1). In practice, the auxiliary gradients computed prior to training can be reused for each target dataset, so the cost is amortized across target datasets. Furthermore, we found that computing gradients for 1000 samples is very fast on a single GPU (~2 minutes per dataset). We will add this clarification to the paper to avoid additional confusion.
>
> >A more minor thing that needs to be taken care of is in line 219. Exploit-only is suppose to be an opposite to Explore-only. So it's confusing how Exploit-only is an extension of Explore-Only. The text later explains the idea, but it needs amendment.
>
> Thank you for pointing out the confusion about Explore-Only vs. Exploit-Only. We’ve altered the text from line 219 to say: “Exploit-Only computes gradient alignment prior to training…” at which point it continues as originally written.
>
> >I did not really see any clear discussion of limitations. While the selection of the datasets are done using the proposed algorithms of balancing exploration and exploitation, there is still an implicit assumption that the datasets are useful. I believe that in language tasks this is less of a problem but in general it could be the case.
>
> You are correct that one of the assumptions for all methods in the Few-shot Learning with Auxiliary Data (FLAD) setting (all prior works included) is that there is at least *some* auxiliary data which will be useful for the target task.
>
> However, one of the main distinctions of our methods from prior works in the FLAD setting is that prior works make a strong assumption that *all* auxiliary data must be useful, and thus the auxiliary datasets should be hand-picked by humans. On the other hand, our methods allow for only a small portion of the auxiliary data to be useful, and we allow the proposed algorithm to explore and find those auxiliary datasets which will be most useful, then exploit them.
>
> Of course, as you mention, because our experiments are on language tasks we cannot make guarantees about any other domains/modalities (e.g. vision, time-series data). However, we believe that our findings suggest that in any domain where multi-task learning is an option, the presented methods can be beneficial.
>
> Thank you for mentioning this limitation. We have added your point, and our thoughts, to the discussion section.

---

> > ### Comment · Reviewer_jiND · 2023-08-11
> >
> > I thank the authors for clarification. I hope  that the author would include the corrections and clarification in the final version, therefore  I increase my ranking from 4 to 6.

---

### Official Review · Reviewer_yskH · 2023-07-05

**Soundness:** 4 excellent
**Presentation:** 4 excellent
**Contribution:** 3 good
**Rating:** 7
**Confidence:** 3

**Summary:**

In the context of Language Models, this paper studies the use of Auxiliary Data during training to gain in generalization. The Few Shot settings is also considered here where proposed tasks only have limited set of labeled data. Compared to previously proposed approaches using auxiliary data, here it is consider to exploit more auxiliary dataset than before. In this context, then it is crucial to select what are the good auxiliary datasets to be used (otherwise performance could degrade) with a reasonable complexity.

A multi-armed bandit (MAB) approach is proposed here to iteratively refine a selection policy among these auxiliary datasets. Two MAB algorithms have been studied (EXP3 and UCB1) in conjunction with different reward functions (based on gradient alignment, gradient magnitude alignment of a combination of both). Experiments are reported on specific training of T5-XL and T0 model. Significant gains with respect to previously proposed approaches are reported. Moreover the complexity associated to the proposed method allows to use high number of auxiliary datasets which helps to boost the performance further.

Discussions on the various aspect of the method are also provided.

**Strengths:**

The paper is clearly written. The adaptation of MAB here allows to derive a performant policy to select auxiliary datasets for training.

Specifically since model updates and policy updates are intrinsically performed, the overall complexity is not armed by the number of auxiliary dataset.

Significant gain are reported with respect to previous works, and relevant discussions are provided upon the importance of rewarding functions used and associated algorithms (e.g. benefits of having exploration and exploitation phases thanks to the bandit).

**Weaknesses:**

As mentioned by the authors, the proposed training has to be performed from scratch for every domain.

**Questions:**

1. Author mentioned that complexity is independent of the number of auxiliary datasets (l. 370-371). Is it really true? When considering even higher number of auxiliary datasets, it could need more iterations to reach a good policy $\pi$ among datasets (need more exploration). So aren't there really an independence there? Or practically since rewards distribution tend to converge quite fast (see fig 4 in annex), impact should be limited?

**Limitations:**

Authors adequately addressed associated limitations.

---

> ### Author Rebuttal · Authors · 2023-08-10
>
> Thank you for the overall supportive review of our work, and for the insightful question about complexity which led us to a very interesting analysis that we now include in the updated draft.
>
> >As mentioned by the authors, the proposed training has to be performed from scratch for every domain.
>
> It’s true that the training needs to be performed from scratch on each target dataset, but this is true of most few-shot methods, and the computational costs of the proposed methods are fairly cheap. We found an average training time of 6 hours on a single 40Gb GPU.
>
> >Author mentioned that complexity is independent of the number of auxiliary datasets (l. 370-371). Is it really true? When considering even higher number of auxiliary datasets, it could need more iterations to reach a good policy ($\pi$) among datasets (need more exploration). So aren't there really an independence there? Or practically since rewards distribution tend to converge quite fast (see fig 4 in annex), impact should be limited?
>
> The computational complexity of a single turn of our method is independent of the number of auxiliary datasets (as you mentioned in Strengths, the policy updates are intrinsically performed). However, you are correct that the computational complexity of the multi-armed bandits are dependent on the number of auxiliary datasets due to their exploration rates (line 146 for Exp3 and 168 for UCB1). Thus, the computational complexity of an entire training run will be influenced by the number of auxiliary datasets, but is challenging to characterize exactly as it relates to the convergence of training a large language model.
>
> To better understand the empirical effects of increasing the number of auxiliary datasets on the time to model convergence, we performed additional analysis of the convergence time for different algorithms. Indeed, we do find that the number of auxiliary datasets requires a longer training, but we find the same to be true for any multitask fine-tuning (see the table below). We find that, compared with baseline methods, our methods require more steps for the smaller auxiliary dataset, but the number of additional steps required to train our MAB-based methods on P3 only increases modestly when increasing the number of auxiliary datasets by a factor of nearly 10 (in contrast with the the explore- and exploit-only baselines, where the number of steps to convergence increases significantly).
>
> *The table below shows the number of training iterations for T0-3B to converge using the training method across the top row, and the auxiliary dataset on the left column. Iterations are averaged over 11 target datasets and 5 seeds leading to 55 experiments averaged per cell.*
>
> | Average # iterations | Explore-Only | Exploit-Only | EXP3-FLAD (GA) | UCB1-FLAD (GA) |
> | ---- | ---- | ---- | ---- | ---- |
> | T0Mix (35 auxiliary datasets) | 570.9 | 549.1 | 769.1 | 700.0 |
> | P3 (260 auxiliary datasets ) | 863.6 | 692.7 | 832.7 | 794.5 |
> | % increase | 51.3% | 26.2 | 8.3% | 13.5% |
>
> We will clarify the statement on lines 370-371 to say “our algorithms have a single-turn computational complexity that is independent of the number of auxiliary datasets”.
>
> We appreciate your thoughtful question! It has led us to some very interesting analysis that we will include in the discussion section of the revised paper.

---

> > ### Comment · Reviewer_yskH · 2023-08-21
> >
> > Thanks for your answers. This clarify my question about complexity.
> >
> > Considering other reviews and feedback provided, I will keep my rating as is.

---

### Official Review · Reviewer_EHmY · 2023-07-06

**Soundness:** 3 good
**Presentation:** 2 fair
**Contribution:** 2 fair
**Rating:** 5
**Confidence:** 4

**Summary:**

This paper focuses on the problem of few-shot learning with auxiliary data (FLAD) problem. The authors formulate FLAD as a multi-armed bandit problem, and employ exploration-exploitation algorithms to tackle it. Experiment results show that the proposed method outperforms exploration or exploitation-only approaches by significant improvement in accuracy.

**Strengths:**

- This paper models the FLAD problem with the multi-armed bandit framework, which makes a difference from previous works. The proposed method scales well to the number of auxiliary datasets, and adds a limited amount of computational and memory overhead.

- The proposed approach shows a significant accuracy improvement over other exploration-only and exploitation-only baselines.

**Weaknesses:**

-The novelty is limited since the proposed multi-armed bandit method is a existing technique.

- The authors should clarify the difference in definitions and settings between FLAD and other similar problems such as general few-shot learning and multi-task learning.

- More experiment results had better be provided. For example, in order to demonstrate the scalability of the proposed method, it would be better to provide what the changes of performance be like with different values of |\mathcal{A}| or |\mathcal{D}_T|.

**Questions:**

Please address the concerns in the Weaknessese.

**Limitations:**

Yes.

---

> ### Author Rebuttal · Authors · 2023-08-10
>
> We thank the reviewer for their useful feedback, which points out some details of the paper that required clarification and for the suggested studies that can strengthen the paper. Our responses are below:
>
> >The novelty is limited since the proposed multi-armed bandit method is a existing technique.
>
> We respectfully disagree with the claim that our work lacks novelty. While our work shows that the MAB framework is a natural fit for FLAD, to the best of our knowledge, MABs have not previously been used in the context of FLAD. The novelty of our work lies in how we adapt and apply multi-armed bandits to FLAD. Our approach results in significantly improved computational complexity compared with prior FLAD methods, allowing us to scale to 100x more auxiliary datasets than previous methods, leading to a 4% accuracy improvement. These results mark a leap in the efficiency and effectiveness of few-shot learning, which can hardly be considered incremental. Thus, while we have built upon existing techniques from multi-armed bandits, we believe the outcome is a transformative contribution to the field of few-shot learning.
>
> >The authors should clarify the difference in definitions and settings between FLAD and other similar problems such as general few-shot learning and multi-task learning.
>
> Thank you for your valuable question. We elaborate on the differences below.
>
> As a reminder, we use the following notation (section 3.1 lines 99-103):
> - $\mathcal{D}_{T}$ is the target task dataset.
> - $\mathcal{D}_{A}$ is the set of all auxiliary datasets.
> - $\mathcal{D}_{a}$ is a single auxiliary dataset.
>
> **Few-shot Learning vs. FLAD**: Both few-shot learning and FLAD are concerned with optimizing model performance on a single target task with a limited number of examples from the target task. In few-shot learning, the model is given only the target task data, $\mathcal{D}\_{T}$, and there is no auxiliary data used (so $\mathcal{D}\_{A}$ is effectively the empty set for few-shot learning). In contrast, in the FLAD setting $|\mathcal{D}\_{A}|$ > 1, and in our settings specifically we use T0Mix and P3 which have $|\mathcal{D}\_{A}| = 35$ and $|\mathcal{D}\_{a}| = 260$ auxiliary datasets respectively. This information all comes from section 4 under the paragraph “Auxiliary datasets'' on lines 206-213.
>
> **Multitask Learning vs. FLAD**: Multitask learning is concerned with optimizing a model for performance on multiple target datasets simultaneously. This is in direct opposition with FLAD which aims to optimize a model for a single target task. Formally, FLAD considers the scenario where $\mathcal{D}\_{A}$ is a set of multiple datasets and $\mathcal{D}\_{T}$ is a single dataset. On the other hand, multitask learning would consider a scenario where $\mathcal{D}\_{A}$ is effectively the empty set, and $\mathcal{D}\_{T}$ is the concatenation of multiple datasets where one attempts to simultaneously optimize for performance all datasets.
>
> We hope this clarifies the differences between few-shot learning, FLAD, and multitask learning. We have made these distinctions clearer in the updated draft.
>
> >More experiment results had better be provided. For example, in order to demonstrate the scalability of the proposed method, it would be better to provide what the changes of performance be like with different values of $|\mathcal{A}|$ or $|\mathcal{D}_T|$.
>
> We thank you for your insightful suggestion of additional ablation studies.
>
> First, we would like to reiterate that the main experiments in this work do include results for two different sizes of  $\mathcal{A}$ (specifically, 35 and 260) to demonstrate the scalability of our proposed methods. On lines 273-277, we discuss how our methods improve between 2.6-4% when increasing $|\mathcal{A}|$ from 35 to 260, while the Loss-Scaling baseline only improves between 0.2-2%. Additionally, in lines 281-285, we show how the Explore-Only and Exploit-Only baselines only improve between 0.7-2%, further showing that our methods demonstrate improved scalability compared with existing FLAD baseline methods.
>
> We agree that the additional ablation studies you suggested will provide for interesting additional analysis. We therefore performed an ablation on values of $|\mathcal{A}|$ in {35, 75, 125, 175, 225, 260} (all taken from P3) for all target tasks using T0 as a base model and repeat the experiments with 3 random seeds. The results are in Figure 1 of the pdf attached to the global rebuttal. We found that the performance for both EXP3-FLAD and UCB1-FLAD jumps significantly from $|\mathcal{A}|$ = 35 to 75, followed by a more steady increase in performance all the way up to $|\mathcal{A}| = 260$. The results show an increase of 2.54 accuracy for EXP3-FLAD and 3.12 for UCB1-FLAD when increasing from 35 to 75 auxiliary datasets, with an additional increase of 1.54 for EXP3-FLAD and 0.47 for UCB1-FLAD when increasing from 75 to 260.
>
> To your point about varying $|\mathcal{D}_T|$, our main experiments do not vary the size of the target dataset in order to maintain fair comparisons with past work that uses the same standard dataset sizes that we did. However, we see that there is value in better understanding the relation of our method with varying target dataset sizes. Unfortunately, we were unable to perform experiments varying $|\mathcal{T}|$ in time for the rebuttal due to computational resources, but will do our best to include them during the discussion period, and will include them in the final draft.
>
> We hope this information is helpful. Please let us know if you have any further questions.

---

> ### Author Response · Authors · 2023-08-16
> **Response Period**
>
> We are pleased to inform you that other reviewers have updated their assessments, and the ratings now indicate an acceptance for our paper. We would like to extend our gratitude for the insights and feedback that you have provided during the review process. We were able to finish a study on varying values of $|\mathcal{D}_{T}|$, but the rebuttal policy suggests that we cannot share the results unless explicitly requested. Should you have any additional questions or concerns, we are available to address them promptly. However, if there are no further inquiries, we kindly ask you to reconsider the update of your rating to acceptance, aligning with the consensus of the other reviewers. Your understanding and collaboration are highly valued, and we believe that aligning the reviews would reflect a cohesive evaluation of our work.

---

### Author Rebuttal · Authors · 2023-08-10

We would like to express our gratitude to the reviewers for their valuable and detailed feedback. We appreciate the time and effort they took to review our draft and provide us with their insightful comments. We carefully consider all of the suggestions and incorporate them into our final version. We are confident that the final product is improved as a result of their feedback.

See the attached pdf for results of a study on adjusting the number of auxiliary datasets, $|\mathcal{A}|\in$ {35, 75, 125, 175, 225, 260}.

---

### Decision · Program_Chairs · 2023-09-21

**Decision:**

Accept (poster)

**Comment:**

The paper tackles the problem of few shot learning with auxiliary datasets (FLAD) by relating it to the multi-armed bandit (MAB) problem. The authors make use of know MAB algorithms in order to build a solution for the FLAD problem. There is a consensus in the reviews that the problem tackled is important, and the experiments show a convincing picture regarding the effectiveness of the proposed method. The main concerns raised were about (1) the novelty of the work, (2) the clarity of manuscript. Regarding (1), the authors indeed did not invent a new algorithm, rather used existing ones as building blocks towards their solution. Given that other papers already tackled the FLAD problem and did not make use MAB in order to solve it, and that the proposed solution here is agreed to perform well, I do not find this to be a major limitation of the paper. For weakness (2), it seems that all the major problems pointed out in the reviews are resolved in the rebuttal. It seems that the required work needed to integrate these clarifications to the paper is not major and can be done towards a camera ready version. Given this, I recommend accepting the paper and urge the authors to carefully go over the discussions here and insert the appropriate changes to their paper.